# Acute exercise rewires the proteomic landscape of human immune cells

David Walzik [1,9], Niklas Joisten [1,9], Alan J. Metcalfe[2,3], Sebastian Proschinger[1], Alexander Schenk[1], Charlotte Wenzel[1], Alessa L. Henneberg [4], Martin Schneider[5], Silvia Calderazzo[6], Andreas Groll [7], Carsten Watzl [8], Christiane A. Opitz [4], Dominic Helm [5] & Philipp Zimmer [1] ✉

The positive effect of exercise on the immune system is widely acknowledged, but the molecular response of immune cells to exercise remains largely unknown. Here, we perform mass-spectrometry-based proteomic analysis on peripheral blood mononuclear cells (PBMC) at a depth of >6000 proteins. Comparing high-intensity interval exercise (HIIE) and moderate-intensity continuous exercise (MICE), matched for time and workload, we identify versatile changes in the proteomic makeup of PBMCs and reveal profound alterations, related to effector function and immune cell activation pathways within one hour following exercise. These changes are more pronounced after HIIE compared to MICE and occur despite identical immune cell mobilization patterns between the two exercise conditions. We further identify an immunoproteomic signature that effectively predicts cardiorespiratory fitness, thus allowing insights into potential exercise-triggered adaptations and immunological health benefits that are mediated by exercise. This study provides a reliable data resource that expands our knowledge on how exercise modulates the immune system, and delivers biological evidence supporting the WHO 2020 guidelines, which highlight exercise intensity as a relevant factor to maintain health.

Physical exercise is one of the most powerful strategies to prevent and counteract acute and chronic diseases across the entire human lifespan. The health benefits of exercise are demonstrated by numerous clinical and observational trials[1,2], but the underlying biological mechanisms are poorly understood. Given the broad implications of the immune system in protecting from disease, there is a clear rationale for comprehensive investigations of the impact of exercise on immune cells.

Both the exercise-induced mobilization of specific immune cells into circulation as well as an increased cytokine release are well-described phenomena in exercise immunology[3]. However, few investigations have evaluated the molecular alterations in immune cells triggered by exercise. Bulk RNA sequencing revealed a complex interplay of up- and downregulated transcripts that peaked two minutes after exercise and returned to baseline within 30–60 min in

[1]Sports Medicine Research Group, Institute for Sport and Sport Science, TU Dortmund University, Dortmund, Germany. [2]Department of Molecular and Cellular Sports Medicine, Institute of Cardiovascular Research and Sports Medicine, German Sport University Cologne, Cologne, Germany. [3]Chest Unit, Centre for Human and Applied Physiological Sciences (CHAPS), Denmark Hill Campus, King's College Hospital, King's College London, London, United Kingdom. [4]German Cancer Research Center (DKFZ), Heidelberg, Division of Metabolic Crosstalk in Cancer and the German Cancer Consortium (DKTK), DKFZ Core Center Heidelberg, Heidelberg, Germany. [5]Proteomics Core Facility, German Cancer Research Center (DKFZ), Heidelberg, Germany. [6]Division of Biostatistics, German Cancer Research Center (DKFZ), Heidelberg, Germany. [7]Department of Statistics, TU Dortmund University, Dortmund, Germany. [8]Leibniz Research Center for Working Environment and Human Factors at TU Dortmund (IfADo), Dortmund, Germany. [9]These authors contributed equally: David Walzik, Niklas Joisten. ✉e-mail: philipp.zimmer@tu-dortmund.de

peripheral blood mononuclear cells (PBMC). Enriched pathways were predominantly related to inflammatory signaling and immune activation, but also included other processes such as cell growth and mobility[4]. While the number and kinetics of transcripts suggest that many expected and some novel signaling pathways are initiated by acute exercise, the proteomic response of PBMCs remains unclear.

Given that the proteomic makeup of cells defines their phenotype and function, bulk proteomic analyses depict a suitable starting point to evaluate the impact of exercise on circulating immune cells. Besides offering a comprehensive overview of the proteomic response of PBMCs to exercise, this also paves the way to an unbiased, data-driven discovery of exercise-induced signaling pathways that may underlie immunological adaptations to exercise. In addition, hypothesis-generating, explorative approaches create a malleable foundation for follow-up investigations applying more targeted strategies with a focus on specific immune cell subsets. In analogy to the World Health Organization (WHO) 2020 guidelines on physical activity, which highlight exercise intensity as a crucial variable for health promotion[5], and considering that high-intensity interval exercise (HIIE) is associated with greater cardiovascular strain compared to moderate-intensity continuous exercise (MICE)[6], we hypothesize that HIIE induces stronger proteomic alterations in PBMCs compared to MICE.

Here, we apply state-of-the-art mass spectrometry-based proteomics and flow cytometry-based immune cell phenotyping in a randomized crossover study to compare how time- and energy-matched HIIE and MICE reshape the immune cell proteome of 23 young healthy adults. We demonstrate that the proteomic makeup of PBMCs is rewired towards immune cell activation and effector function pathways in the recovery phase following exercise and that these changes are indeed more potently induced by HIIE compared to MICE, despite no differences in immune cell mobilization between the two exercise modalities. Using prediction models, we additionally identify an immunoproteomic signature associated with cardiorespiratory fitness, which enables insights into molecular targets that are potentially responsive to exercise training and might thus be involved in immunological health adaptations mediated by exercise. Overall, our data indicate that exercising at higher intensity is necessary to induce proteomic changes associated with immune function, providing biological support for exercise intensity as a central component of the WHO 2020 physical activity guidelines.

## Results

### Study design and participant characteristics

We designed a randomized crossover study comparing time- and workload-matched HIIE and MICE to investigate the impact of exercise intensity on the proteome of immune cells. PBMCs were isolated at baseline, immediately after, and 1 h after each exercise condition (Fig. 1A). In total, data from 23 overnight-fasted recreationally active runners (12 female, 11 male) was collected. Participants exhibited a mean ($\pm$ SD) age of $30 \pm 4$ years, a body mass index (BMI) of $22.2 \pm 2.38$ kg m$^{-2}$, and a cardiorespiratory fitness (measured as peak oxygen uptake, $\dot{V}O_{2peak}$) of $56.64 \pm 6.43$ ml min$^{-1}$kg$^{-1}$ (Table 1 and Supplementary Data S1).

The collected samples were analyzed via state-of-the-art liquid-chromatography mass-spectrometry (LC-MS/MS) and spectral flow cytometry. Comparable to other studies in the field of immunology, our proteomics analysis yielded an excellent coverage with a total of 7385 identified and 6759 quantified proteins. So far, large-scale proteomic analyses on immune cells have mostly been applied in animal models[7] or on resting samples from healthy donors[8], making our study the first to apply these methods in a randomized interventional setting with repeated baselines. After data preprocessing, our dataset comprised 6,039 proteins across 23 participants in 2 exercise conditions with 3 measurement timepoints, respectively. This makes our dataset the largest immune cell proteomics data source available in exercise

context to date (Fig. 1B). Immune cell phenotyping by spectral flow cytometry was performed on a total of 3,537,855 vital lymphocytes (Supplementary Data S2) to assess exercise-induced shifts within the PBMC compartment (Supplementary Fig. S1A). So far, this has been disregarded in exercise studies applying omics approaches on immune cells[4].

### Immune cell mobilization and redistribution is independent of exercise intensity

The mobilization and redistribution of immune cells in response to acute exercise is one of the core phenomena of exercise immunology, and it is nowadays agreed upon that the recovery phase following exercise is characterized by a transmigration of lymphocytes from the bloodstream into peripheral tissues, with crucial implications in many disease settings, including anticancer immunity[9,10], and immunological defense[11,12]. A remaining topic of debate, however, is whether exercise intensity influences the magnitude of immune cell mobilization since previous studies on this topic were matched for exercise duration, but not workload[13,14]. Thus, before dissecting proteomic alterations of PBMCs in response to exercise, we aimed to clarify whether immune cell kinetics differ in dependence on exercise intensity, since this would lead to a different composition of our PBMC samples in response to HIIE and MICE.

By applying unsupervised immune cell clustering using self-organizing maps (SOM), we identified 6 main clusters in our PBMC samples, which were mapped to the corresponding immune cell populations based on their marker expression. Visual inspection of the identified clusters (Fig. 1C) and quantification of exercise-induced cluster shifts resulted in a similar distribution pattern of immune cell clusters between HIIE and MICE with a mean delta of $0.004 \pm 0.9$ % (Fig. 1D and Supplementary Data S2). Confirming these findings, absolute numbers of immune cell populations did not reveal time × condition interaction effects when applying linear mixed models (Supplementary Fig. S1B; Supplementary Data S3) and the proportional contribution of each immune cell population to the PBMC compartment was similar between HIIE and MICE (Fig. 1E and Supplementary Data S4). This suggests that exercise triggers similar mobilization and redistribution patterns independent of exercise intensity and indicates that exercise-induced changes in PBMC composition do not differ between HIIE and MICE.

### Measures of variability indicate high reliability of the generated proteomics dataset

Inter-individual variability of all quantified proteins resulted in median coefficients of variation (CV) of < 5 % for all measurement timepoints and conditions (Fig. 2A). This is considerably lower than in other proteomics studies in exercise context[4,15] and underlines the homogeneity of our study population and the analytic quality of our proteomics pipeline. The applied crossover design additionally enabled us to calculate intra-individual protein variability. The overall mean difference between the two baselines amounted to $0.13 \pm 0.75$% for females and $0.06 \pm 0.59$ % for males (Fig. 2B). To assess variability on a per-protein level, we combined multiple measures of variability (i.e., mean CV at baseline, mean CV in response to exercise, and mean difference between the two baselines) into an integrated variability score. 99.34 % of all quantified proteins revealed a proteomic variability of < 10 % and 83.34 % achieved a score of < 5 % (Fig. 2C and Supplementary Fig.S2A, S2B). In summary, the low variability emphasizes the high quality of our study setup, making our generated proteomics dataset highly reliable.

### Acute exercise alters the immune cell proteome

To obtain first insights into the proteomic alterations induced by exercise, we performed principal component analysis (PCA). Visual inspection of the PCA suggested that the variation within our samples

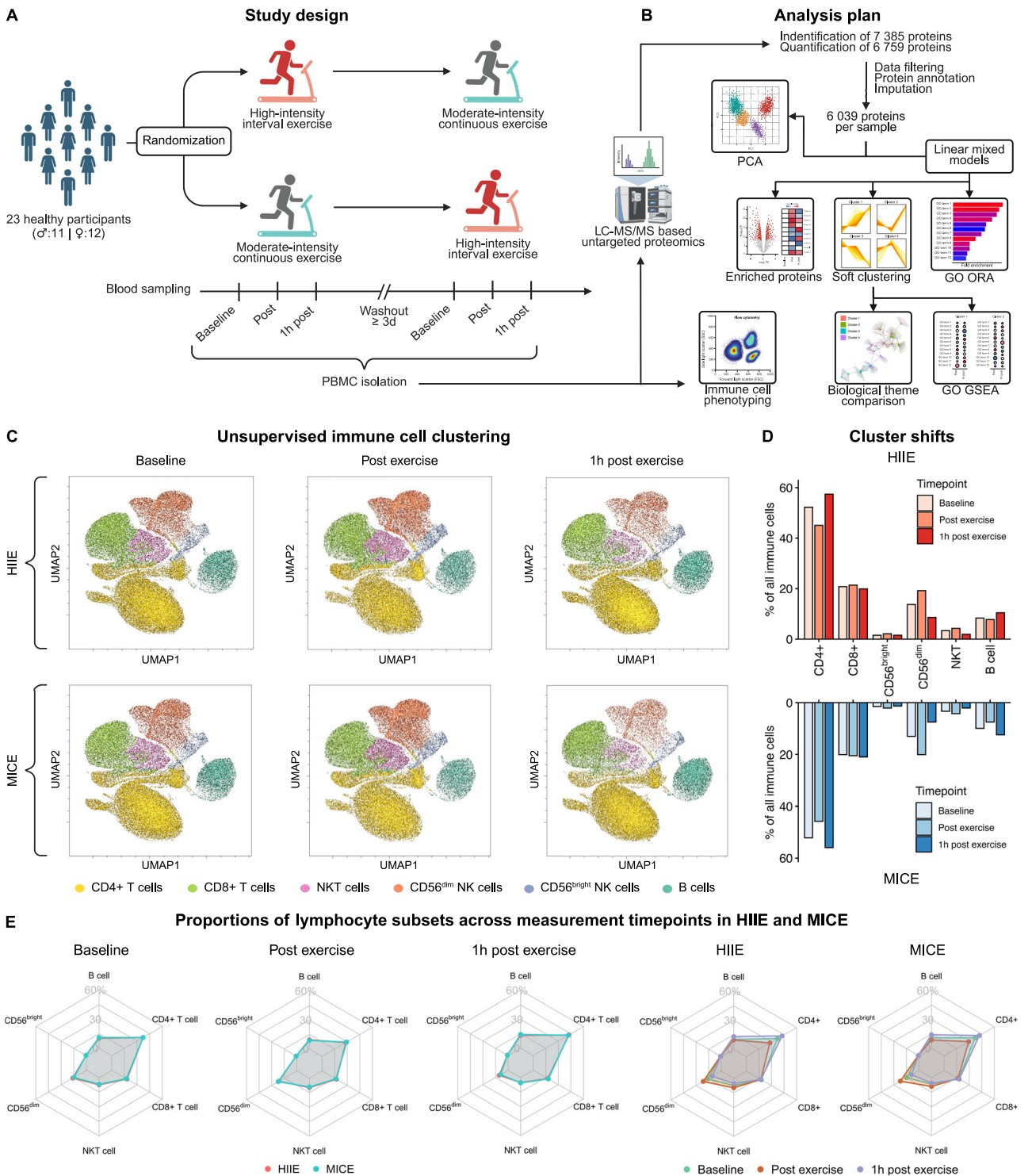

**Fig. 1 | Study design, analysis plan, and exercise-induced immune cell mobilization. A** Overview of the study design, including time- and workload-matched high-intensity interval exercise (HIIE) and moderate-intensity continuous exercise (MICE). **B** Overview of the bioanalytical and bioinformatic methods used to analyze peripheral blood mononuclear cells (PBMCs). **C** Uniform Manifold Approximation and Projection (UMAP) of immune cell clusters identified by unsupervised clustering using self-organizing maps (SOM). Immune cell clusters are displayed color-coded and separated by exercise condition and measurement time point. Each UMAP corresponds to 3000 vital lymphocytes from 22 samples, resulting in a total of 66,000 events. For the UMAP representing 1 h post MICE, only 21 samples were available, resulting in 63,000 events. **D** Comparison of exercise-induced shifts in the identified clusters. **E** Proportions of lymphocyte subsets in response to HIIE and MICE. See also Supplementary Fig. S1 and Supplementary Data S2, S3, and S4. Created in BioRender. Walzik, D. (2025) https://BioRender.com/y85v219.

was mainly accounted for by measurement timepoints but not exercise condition per se (Fig. 2D). PCA also suggested that sex and intervention day had little impact on the variation of our data (Supplementary Fig. S2C). Performing PCA separated by condition and measurement

timepoint indicated that HIIE accounted for more variation 1 h after exercise than MICE (Supplementary Fig. S2D, E).

Next, we compared the impact of HIIE and MICE on proteomic alterations in PBMCs using linear mixed models. We identified 1408

**Table 1 | Overview of participant characteristics**

| | Overall | HIIE-MICE | MICE-HIIE | *p*-value |
|---|---|---|---|---|
| Age [years] | 29.67 ± 4.33 | 30 ± 3.94 | 29.43 ± 4.72 | 0.7577 |
| Height [cm] | 176.96 ± 8.26 | 175.6 ± 6.43 | 177.93 ± 9.47 | 0.5081 |
| Weight [kg] | 69.7 ± 11.17 | 68.83 ± 7.51 | 70.33 ± 13.45 | 0.7538 |
| BMI [kg m$^{-2}$] | 22.2 ± 2.38 | 22.16 ± 1.3 | 22.24 ± 2.98 | 0.9408 |
| $\dot{V}O_{2peak}$ [ml min$^{-1}$ kg$^{-1}$] | 56.64 ± 6.43 | 56.75 ± 7.08 | 56.56 ± 6.19 | 0.9441 |
| HR$_{max}$ [min-1] | 181.43 ± 11.27 | 177 ± 13.07 | 184.85 ± 8.7 | 0.0986 |
| RPE$_{max}$ [A.U.] | 19.38 ± 1.01 | 19.7 ± 0.67 | 19.14 ± 1.17 | 0.1903 |
| RER [A.U.] | 1.07 ± 0.05 | 1.06 ± 0.06 | 1.07 ± 0.04 | 0.5217 |

All data is reported as mean ± standard deviation [minimum; maximum]. A two-sided unpaired *t* test was performed to compare intervention sequences.

time effects, 119 group effects, and 27 time × group interaction effects (Fig. 2E). Including sex as a fixed effect in our analysis did not yield significant results. Dissection of the obtained results revealed more time, group, and time × group interaction effects 1 h after exercise compared to immediately after exercise and more alterations in HIIE compared to MICE (Fig. 2E). In detail, HIIE was marked by 1377 significantly altered proteins, while MICE only caused significant alterations in 64. The fact that immune cell counts and proportions did not differ between HIIE and MICE 1 h after exercise (Fig. 1E and Supplementary S1B) suggests that the proteomic differences are not caused by a distinct PBMC composition. In line with our results obtained by PCA, this suggests that HIIE leads to a more profound reorganization of the PBMC proteome compared to MICE.

**Proteomic alterations differ between HIIE and MICE**

To evaluate proteins that were distinctly regulated by HIIE compared to MICE, we first dissected the interaction effects of our linear mixed models (Supplementary Data S5). Immediately after exercise, we observed 5 proteins with distinct kinetics in HIIE compared to MICE (Fig. 2F). Among these proteins, synaptotagmin-like protein 2 (SYTL2), a crucial contributor to cytotoxic granule exocytosis in lymphocytes[16], displayed a strong increase in response to HIIE, while it remained unaltered in MICE (log$_2$ FC = 1.195, $p = 3.1 \times 10^{-2}$). Similarly, bone marrow stromal antigen 2 (BST2) – known for its role in blocking virus release from infected cells[17] – increased in response to HIIE but decreased in MICE (log$_2$ FC = 1.4, $p = 2.2 \times 10^{-2}$; Fig. 2F, G). This gives first insights into the immunomodulatory potential of HIIE and suggests immunological adaptations dependent on exercise intensity immediately after exercise.

In the recovery phase after exercise, we observed 25 interaction effects. Hierarchical clustering yielded two major clusters of proteins marked by opposed kinetics in HIIE compared to MICE (Fig. 2F, G). For instance, we observed an increase 1 h after HIIE, but a decrease until 1 h after MICE for toll-like receptor 1 (TLR1; log$_2$ FC = 1.06, $p = 1.6 \times 10^{-3}$), BST2 (log$_2$ FC = 1.92, $p = 3.4 \times 10^{-9}$), and cluster of differentiation 302 (CD302; log$_2$ FC = 0.85, $p = 1.5 \times 10^{-2}$). TLR1 is the most abundantly expressed TLR on NK cells[18] and serves as a membrane-bound pattern recognition receptor for microbial lipopeptides that triggers cytokine production and NK cell cytotoxicity upon stimulation[19,20]. Several studies have demonstrated that TLR1 is crucial for antimicrobial defense[21,22], suggesting that exercise-induced increases in TLR1 might reinforce NK cell-mediated immunity against invading pathogens. Of note, BST2 was the only protein that continued to increase from post-exercise to 1 h post exercise in HIIE, suggesting sustained intensity-dependent adaptions in immunological defense.

In contrast, proteins such as SH2 domain-containing protein 1B (SH2D1B), which serves as a cytoplasmic adapter regulating NK cell effector functions[23], or asparagine synthetase (ASNS), which was previously shown to regulate CD8 + T cell activation, differentiation, and effector function[24,25] were marked by a decrease in the recovery period following HIIE, while they remained unaltered or increased in

MICE (Fig. 2F, G; SH2D1B: log$_2$ FC = − 1.25, $p = 1.0 \times 10^{-3}$; ASNS: log$_2$ FC = − 1.25, $p = 7.2 \times 10^{-3}$). Of note, our statistical analysis also yielded several group differences between HIIE and MICE (Fig. 2H, I and Supplementary Data S5).

In summary, our results suggest that the recovery phase following HIIE is marked by more profound alterations of the immune cell proteome compared to MICE. We provide evidence that several proteins related to immune effector function are differentially expressed over time between HIIE and MICE. Against the backdrop of our flow cytometry results, these effects occur despite identical immune cell mobilization patterns between the two exercise conditions.

**Exercise reshapes the immune cell proteome towards effector function**

To add a functional dimension to our results, we made use of the Gene Ontology (GO) Resource[26]. GO over-representation analysis yielded 27 enriched GO terms in HIIE and 9 enriched GO terms in MICE. Interestingly, enriched GO terms were centered around immune effector functions in both HIIE and MICE, with biological processes like "disruption of cell in another organism" or "killing of cells of another organisms" yielding high enrichment (Fig. 3A).

For proteins altered by MICE, GO over-representation analysis additionally yielded several biological processes related to lymphocyte effector function, such as "lymphocyte mediated immunity" or "T cell mediated immunity". Given that the over-representation analysis was conducted with much more proteins for HIIE, we additionally identified several cellular components and molecular functions in this analysis. Semantic evaluation of the identified GO terms underlined their association with immune effector function. For instance, "exogenous protein binding", "virus receptor activity", and "endopeptidase activity" are known molecular functions in the context of immunological defense against viruses[27,28]. Similarly, "proteasome core complex" and "peroxisomal membrane" depict cellular components associated with such molecular function and biological processes (Fig. 3A). Collectively, our GO over-representation analysis points towards enhanced regulation of immune effector functions in response to both HIIE and MICE (Supplementary Data S6).

**Time-resolved protein changes differ between HIIE and MICE**

Within all proteins altered by exercise ($n = 1408$), we found 1344 proteins that were uniquely altered by HIIE, 31 proteins that were uniquely altered by MICE, and 33 proteins that were altered by both exercise conditions (Fig. 3B). Analysis of proteins altered by HIIE suggested two major protein clusters that were characterized by increased or decreased protein abundance 1 h after HIIE compared to baseline (Fig. 3C). Considering the large number of proteins altered by HIIE compared to MICE, we took different approaches in analyzing the time effects of each exercise condition.

For proteins altered by both exercise conditions and proteins uniquely altered by MICE, we performed hierarchical clustering to

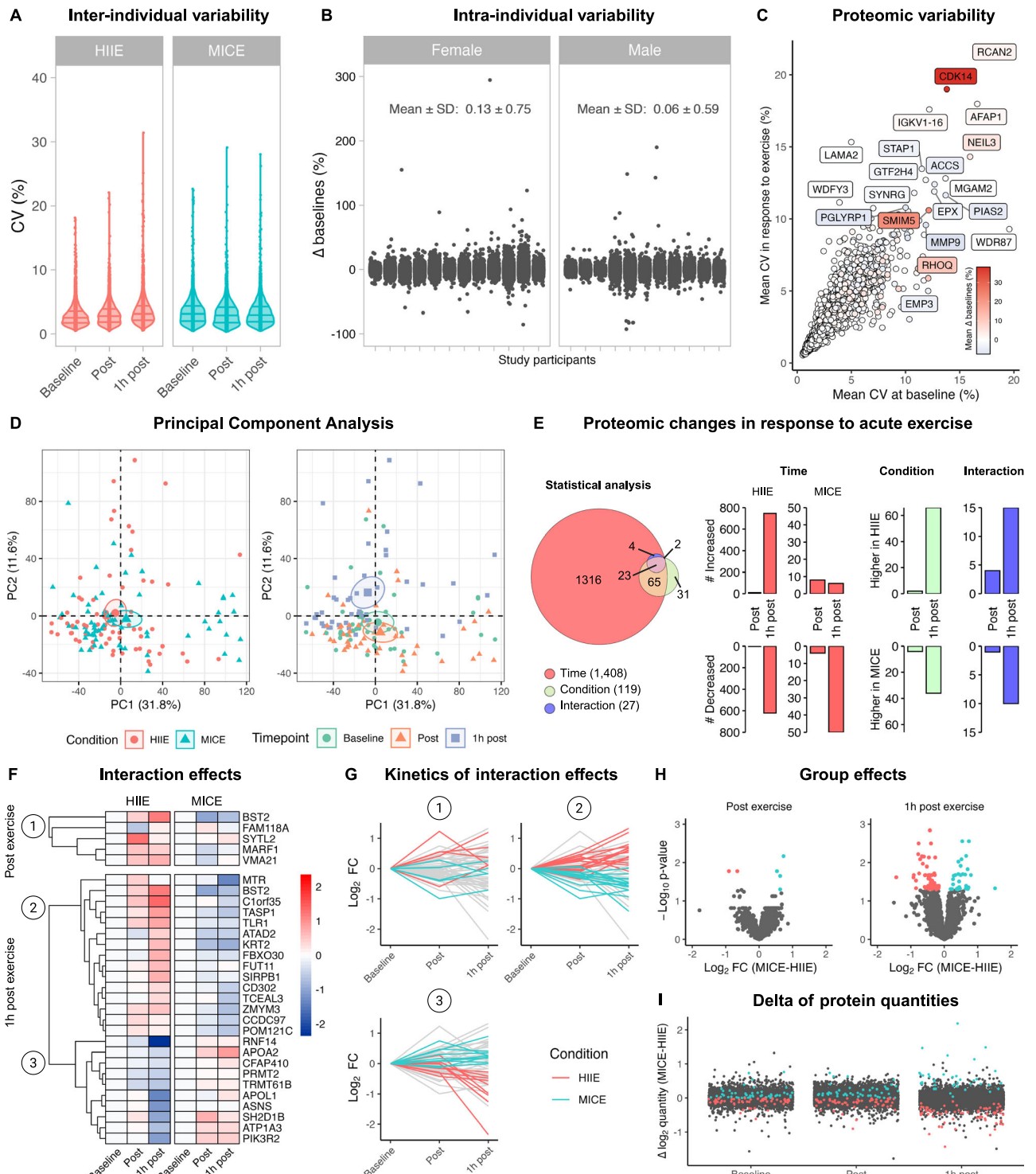

**Fig. 2 | Measures of variability and proteomic changes in response to exercise.**
**A** Inter-individual variability of all quantified proteins expressed as coefficients of variation (CVs) separated by exercise condition and timepoint. **B** Intra-individual variability of all quantified proteins expressed as delta between the two baselines separated by sex and study participant. **C** Proteomic variability of all quantified proteins. **D** Principal component analysis of all samples using exercise condition and measurement timepoint as metadata. Small symbols indicate individual samples. Big symbols and circles indicate the mean and 95 % confidence interval of the corresponding data subset. **E** Quantification of proteomic changes in response to exercise. Linear mixed models containing exercise condition, measurement timepoint, and the interaction between both as fixed factors were applied ($N = 23$). **F** Interaction effects between time and exercise condition for HIIE and MICE. Dendrograms depict clusters identified by full-linkage hierarchical clustering. **G** Kinetics of proteins displaying interaction effects separated by the identified clusters (1–3). **H** Group differences between HIIE and MICE. Significant proteins are colored by exercise condition. **I** Delta of protein quantities between HIIE and MICE. Proteins displaying significant group differences are colored by exercise condition. See also Supplementary Fig. S2 and Supplementary Data S5.

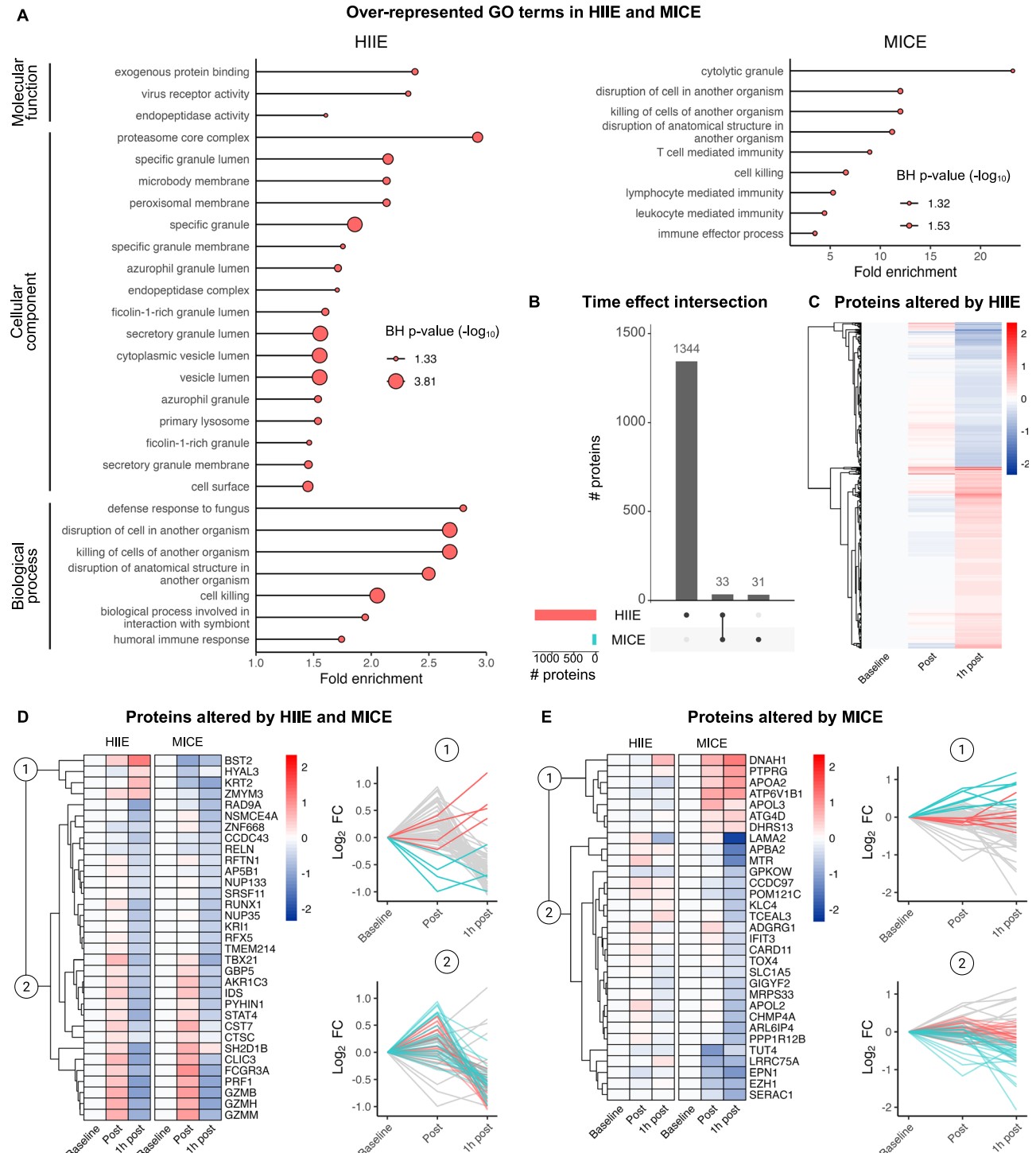

**Fig. 3 | Exercise reshapes the immune cell proteome towards effector function and causes distinct alterations in protein abundance in response to HIIE and MICE. A** Gene Ontology (GO) over-representation analysis comparing significantly altered proteins in HIIE ($n = 1377$) and MICE ($n = 64$) with all proteins quantified in this study ($n = 6039$). **B** Overview of proteins altered by HIIE, MICE, or both exercise conditions. **C** Overview of proteins uniquely altered by HIIE.

**D** Overview of proteins altered by both exercise conditions. The first two clusters identified by hierarchical clustering separate proteins displaying different kinetics in HIIE and MICE (1) from proteins displaying similar kinetics (2). **E** Overview of proteins uniquely altered by MICE. The first two clusters identified by hierarchical clustering separate proteins increasing in response to MICE (1) from proteins decreasing in response to MICE (2). See also Supplementary Data S5 and S6.

identify proteins displaying similar kinetics. Interestingly, when analyzing proteins that were altered by both exercise conditions, hierarchical clustering yielded 2 major protein clusters: one cluster containing proteins with similar kinetics between HIIE and MICE and one cluster containing proteins with different kinetics (Fig. 3D). In absolute terms, most of the proteins responded similarly with only 4

proteins showing higher values in HIIE, including the antiviral protein BST2, which we previously identified in our analysis of time × group interaction effects (Fig. 2F). In addition, many of the proteins that were shared between HIIE and MICE were associated with immune effector functions, suggesting a shared regulation of several immunological processes by exercise. Examples of such proteins include

granzymes (e.g., GZMB, GZMH, GZMM), perforin-1 (PRF1), or guanylate-binding proteins 5 (GBP5; Fig. 3D).

Similarly, hierarchical clustering of proteins uniquely altered by MICE identified 2 major clusters that separated proteins that decreased in response to MICE from proteins that increased, while showing no alterations in HIIE, respectively (Fig. 3E). In line with the observed time × group interaction effects (Fig. 2F) most proteins displayed lower abundance in response to MICE (Fig. 3E). Of note, although the lower number of time effects and the decreased abundance of many proteins might suggest reduced immune effector functions in response to MICE, it is crucial to emphasize that several proteins with immunological functions, especially those jointly regulated between HIIE and MICE, revealed increased abundance in response to MICE as well. Thus, while our data suggests that the immunoproteomic impact of MICE seems to be less pronounced than that of HIIE, there is no conclusive evidence suggesting reduced immune effector function per se in response to MICE.

### Identification of shared and unique immune effector functions regulated by HIIE

To dissect the proteomic alterations in response to HIIE, we performed fuzzy c-means clustering and mapped the altered proteins ($n = 1377$) to four distinct clusters by means of their relative membership (Fig. 4A and Supplementary Data S7). The four identified clusters confirmed what hierarchical clustering had previously suggested, i.e., two major clusters marked by increased or decreased protein abundance 1 h after HIIE (Figs. 3C and 4A). We next leveraged biological theme comparisons[29] and identified 576 biological processes, 132 molecular functions, and 187 cellular components associated with the proteins altered by HIIE (Supplementary Data S8). By generating gene-concept networks of the five most significant GO terms in each ontology, we observed both shared and unique GO terms across our four protein clusters (Supplementary Fig. S3A–C).

To quantify functional differences and similarities between the identified protein clusters, we performed gene set enrichment analyses (GSEA)[30] and observed a total of 169 enriched GO terms (Fig. 4B and Supplementary Data S9). Interestingly, cluster 4 did not yield any enriched GO terms and re-evaluation of the underlying statistics demonstrated that the individual GO terms did not reach the significance threshold. These findings were validated using the STRING resource[31]. We then focused our attention on GO terms that were shared across protein clusters 1 – 3 and obtained 11 shared biological processes. Semantic evaluation confirmed their close connection to immune function, as exemplified by GO terms like "cell killing", "leukocyte activation", or "defense response" (Fig. 4B). Analysis of the underlying proteins resulted in a core proteome consisting of 369 proteins, most of which changed in abundance in the recovery phase following HIIE (Fig. 4C). This suggests that the biological processes regulated by HIIE are driven by proteomic alterations in the recovery phase. We observed similar results for the 27 GO terms that were shared between clusters 1 and 2, and the 15 GO terms that were shared between clusters 2 and 3 (Fig. 4B and Supplementary Fig. S4A–C).

Concerning GO terms that were uniquely enriched in a specific protein cluster, we identified 29 GO terms uniquely enriched in cluster 1, 62 GO terms uniquely enriched in cluster 2, and 25 GO terms uniquely enriched in cluster 3 (Fig. 4B, D). Among the GO terms enriched in cluster 1 we found enriched regulation of "endopeptidase activity" and "cellular response to organic substance" (Fig. 4D). Similarly, cluster 2 demonstrated enriched regulation of "glycosaminoglycan binding" and "cell migration", which are crucial processes in the context of exercise-induced transmigration of immune cells from the bloodstream into peripheral tissues. In contrast, cluster 3 was characterized by a decreased regulation of several cellular components and biological processes 1 h after exercise, which can be attributed to the underlying protein kinetic (Fig. 4A). In summary, GSEA

suggested a profound regulation of immune effector processes in the recovery phase following HIIE, which were in part shared and in part unique for specific protein kinetics.

### Identification of an immunoproteomic signature associated with cardiorespiratory fitness

We ultimately leveraged our generated dataset to explore potential long-term immune adaptations to exercise training since this depicts a crucial starting point in understanding the immune-mediated health benefits triggered by exercise. Taking a data-driven approach, we started by pooling the baseline data of all our analyses, including participant characteristics as well as flow cytometry and LC-MS/MS results. This comprehensive dataset was then used to investigate potential pairwise associations with $\dot{V}O_{2peak}$, a gold standard marker of cardiorespiratory fitness that is highly responsive to exercise training. In a first step, we calculated Spearman's rank correlation coefficients ($r_S$) and preselected features that displayed moderate to high correlation ($r_S > 0.4$ or $< -0.4$) with $\dot{V}O_{2peak}$. This resulted in a reduction of our dataset from 6063 to 260 features (Supplementary Data S10).

To establish an elaborate connection between these features and cardiorespiratory fitness, we next performed prediction analyses. Ridge regression yielded an R-squared of 0.61 and a mean squared error (MSE) of 14.1 and visual inspection of the ranked coefficients revealed a homogeneous distribution of features with positive or negative impact on $\dot{V}O_{2peak}$ prediction, respectively (Fig. 5A). We next focused on the 20 features with the highest positive or negative impact on $\dot{V}O_{2peak}$ prediction and defined these as immunoproteomic signature predictive of cardiorespiratory fitness (Fig. 5B). Interestingly, among these proteins, nicotinamide phosphoribosyltransferase (NAMPT), a key enzyme of nicotinamide adenine dinucleotide (NAD$^+$) metabolism, demonstrated the highest positive impact on $\dot{V}O_{2peak}$ prediction. NAMPT plays a crucial role in salvaging intracellular NAD$^+$ and was previously shown to be exercise-responsive in skeletal muscle[32–34], but also other target tissues like immune cells[35–37]. In addition, several studies have suggested a direct antiviral function of NAMPT in host defense[38,39]. Our results support this notion and suggest that repeated exposure to exercise, which results in greater cardiorespiratory fitness, equips immune cells with a higher metabolic capacity, thereby linking to the immune effector functions previously identified in this work. Similarly, we observed a positive impact on $\dot{V}O_{2peak}$ prediction for succinate receptor 1 (SUCNR1), a G-protein coupled receptor that was previously shown to control exercise capacity and systemic glucose homeostasis in mice[40,41].

Besides features with positive impact on $\dot{V}O_{2peak}$ prediction, our immunoproteomic signature also contained several proteins with a negative impact (Fig. 5B). Among these features, phosphatidylserine decarboxylase (PISD), an enzyme involved in lipid droplet biogenesis[42], and caspase recruitment domain family, member 8 (CARD8), a pattern recognition receptor that regulates inflammasome activation and production of pro-inflammatory cytokines[43], stood out due to their involvement in metabolism and inflammation. Interestingly, we also observed various inter-feature correlations within our immunoproteomic signature (Fig. 5C).

To confirm that our immunoproteomic signature was associated with cardiorespiratory fitness, we next calculated individual immunoproteomic signature scores using the mean beta coefficients of the ridge regression and individual protein abundancies (see methods for details). Individual scores yielded a significant positive correlation with $\dot{V}O_{2peak}$ ($R = 0.84$, $p = 5.1 \times 10^{-7}$) and were higher in participants with greater cardiorespiratory fitness based on a median split performed on our study cohort ($p = 0.00068$, Fig. 5D). Similarly, NAMPT protein levels displayed a positive correlation with $\dot{V}O_{2peak}$ ($R = 0.54$, $p = 0.0073$, Fig. 5E). This confirms that the immunoproteomic signature and underlying proteins such as NAMPT, are associated with cardiorespiratory fitness.

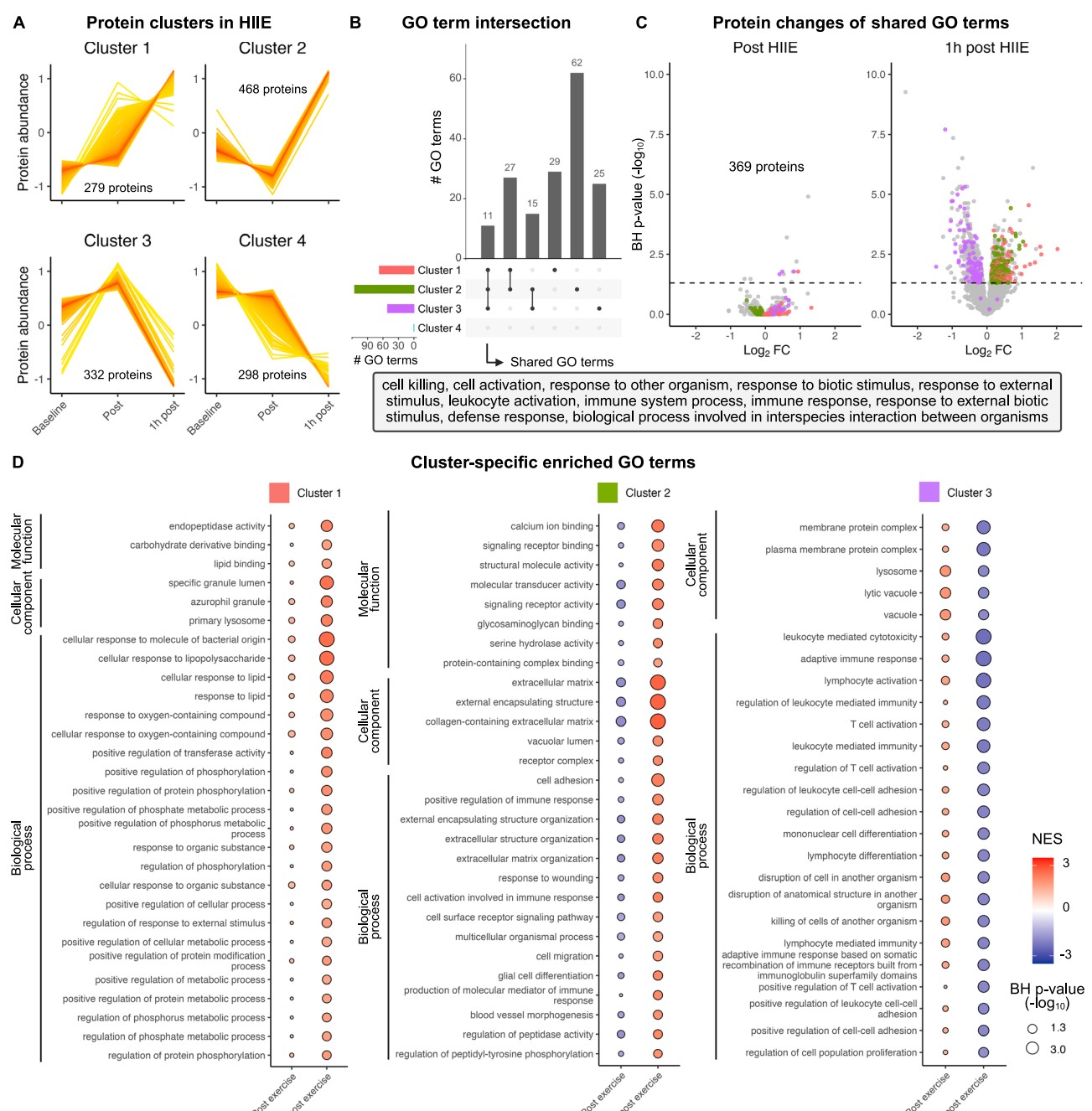

**Fig. 4 | Identification of shared and cluster-specific immune effector functions in the recovery phase following HIIE.** **A** Protein clusters identified by fuzzy c-means clustering in response to HIIE. **B** Shared and unique GO terms across protein clusters 1–3. **C** Identification of core biological processes shared across clusters 1–3 and temporal regulation of the underlying proteins. **D** Cluster-specific GO terms regulated by HIIE. See also Supplementary Fig. S3 and S4, and Supplementary Data S9.

Having identified this signature, we ultimately aimed to investigate vice versa, whether cardiorespiratory fitness also explains most of the variance in the immunoproteomic signature and the underlying proteins. To assess this, we applied a variance partitioning approach, in which we compared the variance explained by $\dot{V}O_{2peak}$ to the variance explained by other participant characteristics such as age, sex, BMI, and participant ID. These comparisons were performed across three different feature sets: a feature set containing all flow cytometry and LC-MS/MS data, a feature set containing the preselected features for $\dot{V}O_{2peak}$ prediction (Fig. 5A), and a feature set containing the immunoproteomic signature (Fig. 5B). While participant ID accounted for significantly more variance than $\dot{V}O_{2peak}$ in the overall feature set and

the feature set containing the preselected features, this difference was no longer apparent in the immunoproteomic signature (Fig. 5F). In addition, in both, the preselected feature set and the immunoproteomic signature, $\dot{V}O_{2peak}$ explained significantly more variance than age, sex, or BMI (Fig. 5F and Supplementary Fig. S5A). This demonstrates that cardiorespiratory fitness is closely related to the immunoproteomic signature.

We next dissected the variance explained by participant characteristics for each protein contained in the immunoproteomic signature to evaluate which of the proteins are most strongly influenced by cardiorespiratory fitness. Interestingly, $\dot{V}O_{2peak}$ explained 13.62 % of the total variance (Supplementary Fig. S5B) and 93.66 % of the

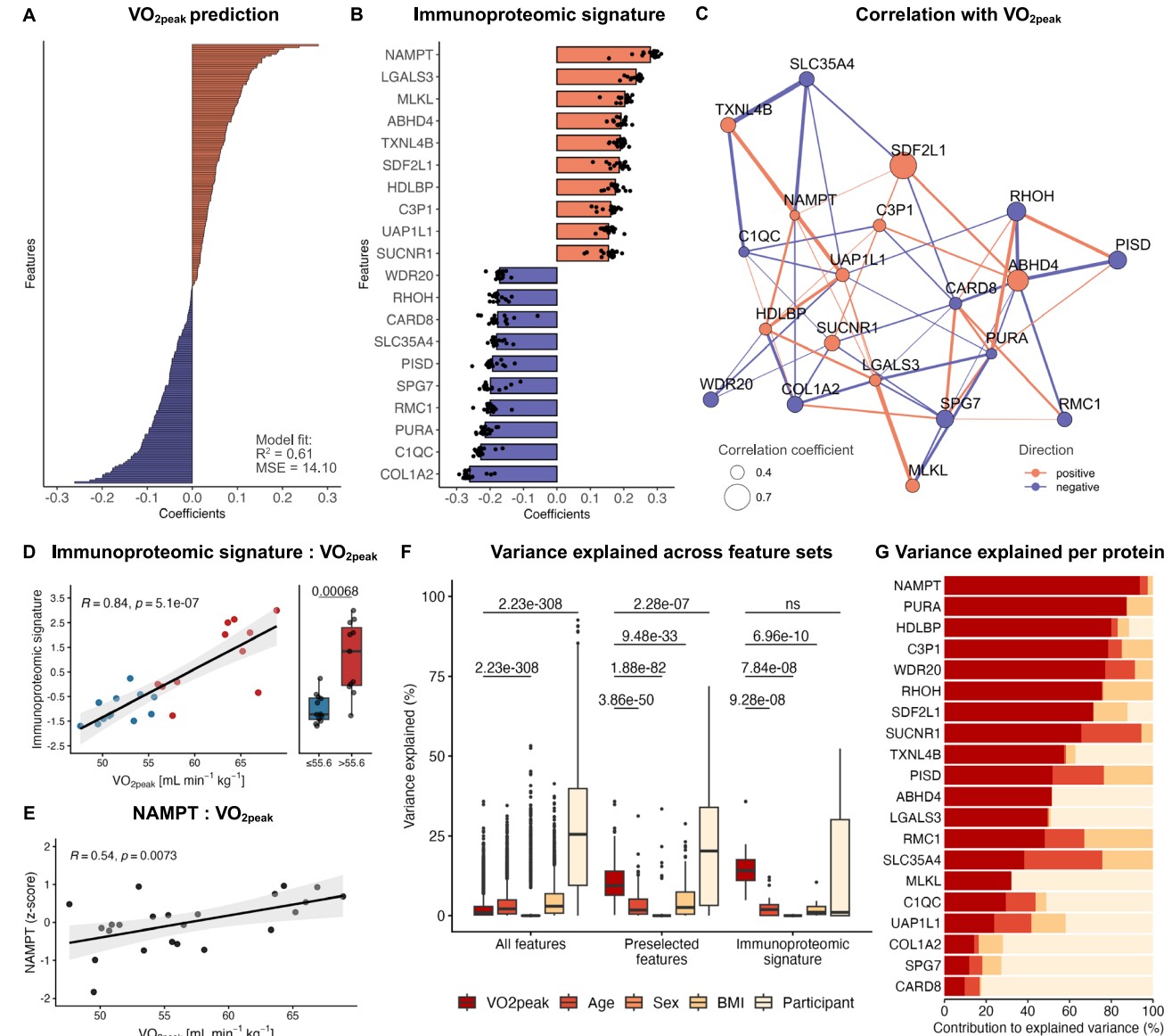

**Fig. 5 | Identification of an immunoproteomic signature associated with cardiorespiratory fitness. A** Ridge regression coefficients of preselected features ($r_S$ $\dot{V}O_{2peak}$ > 0.4 or < − 0.4) used for $\dot{V}O_{2peak}$ prediction. **B** Ridge regression coefficients of the 20 proteins with the highest predictive power for $\dot{V}O_{2peak}$. These proteins were defined as immunoproteomic signature (N = 21). **C** Correlation network of the 20 proteins included in the immunoproteomic signature. Dot sizes and colors represent the strength and direction of correlation with $\dot{V}O_{2peak}$. Connection width and colors represent the strength and direction of correlation between proteins. Connections are displayed for $r_S$ > 0.3 or < − 0.3. **D** Immunoproteomic signature scores correlate with $\dot{V}O_{2peak}$ and differ between cardiorespiratory fitness levels. Boxplots are based on a $\dot{V}O_{2peak}$ median split of our cohort (median $\dot{V}O_{2peak}$ = 55.6 ml min⁻¹ kg⁻¹). Welch's two-sample t-tests were conducted to compare immunoproteomic signature scores between cardiorespiratory fitness groups

(N = 23). **E** NAMPT protein levels correlate with cardiorespiratory fitness (N = 23). **F** Variance explained by participant characteristics across different feature sets. Two-sided pairwise Wilcoxon rank-sum tests with Bonferroni-correction were conducted to compare the variance explained by each participant characteristic to the variance explained by $\dot{V}O_{2peak}$ (N = 21). Significant results are displayed for participant ID vs. $\dot{V}O_{2peak}$ and when $\dot{V}O_{2peak}$ yielded significantly higher results than other participant characteristics. ns, not significant. **G** Contribution of participant characteristics to the explained variance for each protein of the immunoproteomic signature (N = 23). Gray shading of the regression lines represents the 95% confidence interval. Boxplots show the median (center line), the 25th and 75th percentiles (bounds of the box), and whiskers extending to 1.5 × the interquartile range. Points beyond the whiskers represent outliers. See also Supplementary Fig. S5 and Supplementary Data S10.

explained variance within NAMPT (Fig. 5F). This rendered NAMPT the protein with the highest relative contribution of $\dot{V}O_{2peak}$ to the total amount of explained variance, thus confirming our previous finding that PBMC NAMPT levels are dependent on cardiorespiratory fitness.

Collectively, our results suggest that the identified immunoproteomic signature is strongly related to cardiorespiratory fitness and that vice versa, cardiorespiratory fitness also explains most of the variance in the immunoproteomic signature. In addition, many of the underlying proteins, such as NAMPT, are dependent on cardiorespiratory fitness. This suggests a potential involvement of these

proteins in reshaping the phenotype of immune cells in response to exercise training. In a broader context, these findings could serve as a molecular foundation for immunological health benefits mediated by regular exercise.

## Discussion

A better understanding of the molecular underpinnings of physical exercise is needed to individualize exercise training recommendations and maximize their efficacy in mediating health benefits. While some human studies have addressed exercise responses in skeletal muscle

and blood plasma using state-of-the-art systems biology approaches[44–48], the impact of exercise on the immune system is less well understood. Here, we provide a comprehensive resource on how two different aerobic exercise stimuli rewire the proteomic makeup of PBMCs. We applied a robust randomized crossover design, including two standardized baseline measurements in a large sample size for proteomic analyses in humans. Our findings expand the literature by >1000 proteomic changes in response to acute exercise in PBMCs. Particularly, immune effector function and cell activation pathways are regulated, and higher intensity is needed to stimulate these changes. Finally, we demonstrate that baseline proteomics data can predict $\dot{V}O_{2peak}$ and identify potential exercise-responsive targets in PBMCs that warrant further investigation as immunological mediators of health adaptations triggered by exercise.

The acute exercise-induced mobilization of effector cells like NK cells and cytotoxic (CD8+) T cells is well-investigated[3,49] but less is known on changes in their proteomic makeup and the resulting cell functions. We identified comprehensive alterations in the immune cell proteome associated with cell function and activation pathways that match previous studies evaluating functional outcomes in different effector populations[50–52]. The regulation of immune effector functions suggests a transient state of immunomodulation following acute bouts of exercise, thereby elucidating the mechanisms of action underlying the benefits of exercise training for disease prevention. The present findings support the concept that acute exercise, particularly at higher intensities, can boost immune responses to pathogens, which stands in contrast to an alternative concept of immunosuppressive states following acute exercise[53]. The acute exercise-induced activation of immune effector pathways such as "cell killing", "leukocyte activation", or "defense response" suggests a transient state of elevated immune surveillance, which may be specifically beneficial for the prevention of infectious diseases or cancer.

Furthermore, there is the general premise that acute exercise stimulates immune system components, while chronic exercise training mediates anti-inflammatory effects[54]. Longer-term anti-inflammatory effects driven by immune cell adaptations, e.g., through the differentiation of regulatory T-cells[55], may be induced in response to each single bout of acute exercise. Given that we observed profound proteomic alterations in the recovery phase following HIIE, future investigations are needed with longer-term follow-up periods to not only detect transient acute effects but also focus on more persistent adaptations in immune cell subsets. The associations between the PBMC proteome and $\dot{V}O_{2peak}$ further support this premise and indicate that exercise-induced immunoregulation may also be beneficial for chronic diseases such as type 2 diabetes or obesity, particularly when considering long-term anti-inflammatory effects. In this context, an interesting finding of our analysis is that proteins associated with mitochondrial processes were decreased following HIIE (Supplementary Fig. S3), however, baseline $\dot{V}O_{2peak}$ prediction yielded several proteins associated with increased metabolic/functional capacity (Fig. 5B). Overall, the mobilization and redistribution of immune cells with exercise and the parallel activation of different cellular pathways, depicts a promising physiological framework for future studies on the health-promoting effects of exercise mediated by immune cells, especially considering their mobile nature and versatile recruitment to different peripheral tissues[54].

Interestingly, our results indicate that the observed changes in the proteomic makeup of PBMCs occur independent of exercise-induced mobilization and redistribution of immune cells. This is demonstrated by the fact that we observed far more proteomic alterations in response to HIIE compared to MICE, although the underlying PBMC composition did not differ between the two exercise conditions. Although previous investigations have neglected this crucial component of exercise immunology, our proteomics results are temporally in line with transcriptomic alterations identified before[4]. In this context,

our open source web application, which can be found at https://sportsmedicine-dortmund.shinyapps.io/beat, offers the opportunity to mine the underlying dataset for specific proteins of interest, thus informing new hypothesis-driven research in the field of exercise immunology.

Our results support the WHO recommendations on physical activity, which highlight the superior role of high exercise intensity for health promotion[5]. From an immunological perspective, we found distinct responses of HIIE and MICE when matching the interventions for duration and workload, and thus conclude that exercising at higher intensity is crucial to induce more profound changes in the PBMC proteome. This might serve as a potential biological foundation for a recent comprehensive analysis revealing a superior effect of exercise intensity versus volume on longevity at a population-based level[56].

Finally, while previous work has elucidated the molecular underpinnings of cardiorespiratory fitness[4,46,48], a possible link to immune cells has not yet been explored. We observed strong associations with $\dot{V}O_{2peak}$ for several proteins, including NAMPT, which is crucial for cellular energy metabolism. Confirming these findings, we have recently demonstrated that NAMPT expression of human PBMCs increases in response to acute exercise[36]. Overall, this suggests an interrelation between acute exercise stimuli, immunometabolic competence, and cardiorespiratory fitness and suggests a putative role of PBMCs as a peripheral mirror for systemic health.

Our work has some important limitations. One main limitation is the lack of cytokine profiling and additional omics layers, such as transcriptomic or metabolomic data. These data would have provided a more holistic systems biology perspective and should be considered in future investigations. In addition, the absence of functional assays prevents clear conclusions on immune cell function in response to acute exercise. Importantly, the generalizability of the findings is limited to young, healthy, and trained individuals, as the study investigated a relatively homogeneous cohort. In this cohort, HIIE was required to induce pronounced changes in PBMC effector pathways. However, these findings may be influenced by age- or training-related effects. It remains unclear whether MICE would be sufficient to elicit similar proteomic responses in PBMCs in older, sedentary, untrained, or diseased populations.

Future investigations are warranted to address these limitations to better understand and apply the findings of the present study, also in other cohorts with different age groups and fitness levels. Specifically, follow-up studies using sorted immune cell subsets, and particularly effector immune cells, for multi-omics analysis will deepen our understanding of how exercise modulates immune cells from a systems biology perspective. Applying cytotoxicity, cytokine secretion, differentiation and proliferation assays will add important knowledge on the functional consequences of these immune cell subsets.

In conclusion, we identified >1000 exercise-induced alterations in the PBMC proteome and provide a valuable data resource for future research. The identified changes were particularly related to immune effector function, serving as a mechanistic link for the preventive and therapeutic impact of regular exercise. In line with the WHO 2020 guidelines on physical activity, acute exercise at higher intensity elicited greater changes in the regulation of cell function and activation pathways, providing supportive biological evidence for the relevance of exercise intensity as an important factor when planning and structuring exercise training programs for health promotion. Finally, the associations between the PBMC proteome and $\dot{V}O_{2peak}$ shed light on potential molecular mediators of immunological health.

## Methods

### Participant recruitment and informed consent
Prior to enrollment of the first participants, the study received ethical approval by the local ethics committee of the German Sport University Cologne, which works according to the World Medical Association's

Declaration of Helsinki. The study meets the National Institutes of Health definition of a clinical trial and was prospectively registered in the German Clinical Trials Register (DRKS00017686). Study eligibility was assessed for 28 healthy adults aged between 18 and 35. To ensure complication-free execution of the high-intensity interval exercise on the treadmill, participants required a weekly running volume of 2–5 h and a body mass index < 30. Any previous medical history of muscle disorders, cardiac or kidney diseases, as well as regular intake of medication or nutritional supplements, were treated as exclusion criteria. For female participants, breast-feeding or an ongoing pregnancy were also treated as exclusion criteria. Of the 28 subjects assessed for eligibility, two were considered ineligible due to acute infections. The remaining 26 participants provided written informed consent and were enrolled in the study. After baseline testing, two further participants dropped out due to orthopedic problems while running (Achilles injuries). For one participant, the biomaterial did not suffice to run analyses, which resulted in a total of 23 participants. An overview of all participant characteristics is displayed in Table 1.

## Study design

Participants enrolled in this randomized crossover study were scheduled for three visits to an exercise physiology laboratory of the German Sport University Cologne: Baseline testing, a HIIE session, and a MICE session. For each visit, participants were asked to arrive overnight-fasted and refrain from alcohol and caffein intake in the 24 h prior. Water intake was permitted ad libitum. All visits were scheduled between 07:00 and 10:00 am to account for a potential circadian impact on performance and biological outcomes. The minimum timeframe between each of the three visits was 72 h, to prevent potential carryover effects.

## Baseline testing

During baseline testing, written informed consent was obtained from participants and demographic and anthropometric characteristics were recorded. Afterwards, participants underwent cardiopulmonary exercise testing.

**Cardiopulmonary exercise test (CPET).** To standardize the exercise intensity between participants for the HIIE and MICE sessions, respectively, cardiorespiratory fitness was assessed as peak oxygen consumption ($\dot{V}O_{2peak}$) in an incremental CPET during baseline testing. The CPET was performed on a motorized treadmill (Woodway ELG 90, Weil am Rhein, Germany) that was set to 1 % incline for all sessions. The warm-up consisted of 5 min at 6–8 km h$^{-1}$. Afterwards, participants began running at 8 km h$^{-1}$. The speed of the treadmill was then increased by 1 km h$^{-1}$ every 60 s until participants reached volitional exhaustion. During the test, heart rate was recorded continuously (Polar FS1C, Kempele, Finland), and rate of perceived exertion was recorded prior to each increase in intensity. Participants were verbally encouraged to continue running by the supervising researcher. After reaching volitional exertion, participants were given a 5 min break before taking up exercise again for a $\dot{V}O_{2peak}$ verification test. For this test, the speed of the treadmill was set 1 km h$^{-1}$ higher than what the participants had finished with. Just before the verification test, participants ran for 3 min at 8 km h$^{-1}$. The speed was then increased to the target speed within 20 seconds, and participants were instructed to run as long as possible. During the entire CPET, participants wore a face mask that was connected to a spirometer (Cortex Metalyzer 3B, CORTEX Biophysik GmbH, Leipzig, Germany) to collect breathing gases breath-by-breath. The highest 15 s interval during the CPET was used to calculate $\dot{V}O_{2peak}$.

## Randomization

To prevent sequence effects arising from the order in which HIIE and MICE were conducted, participants were randomized into one of two exercise intervention sequences after baseline testing: HIIE-MICE or MICE-HIIE. Following the minimization procedure by Pocock and Simon[57], randomization was performed via concealed allocation (1:1) using the software Randomization in Treatment Arms (RITA; Evidat, Lübeck, Germany). Age, BMI, and $\dot{V}O_{2peak}$ were used as stratification factors. The intervention sequences did not differ in terms of participant characteristics, indicating that our randomization was unbiased (Table 1).

## Exercise interventions

Exercise intensities for the HIIE and MICE sessions were calculated as percent of $\dot{V}O_{2peak}$ for each participant to ensure that all participants exercised at the same intensity. The exercise protocols for HIIE and MICE were designed in a time- and workload-matched manner as previously described[58,59] to isolate exercise intensity as the only differing variable between the two exercise conditions. This time- and workload-matched design is crucial to draw unbiased conclusions on the impact of exercise intensity. In addition, we chose these exercise protocols because a duration of 50 min represents a conventional length for acute exercise bouts across different settings (e.g., performance- or health-oriented contexts). Both exercise sessions were performed on the same treadmill that was also used for the CPET at baseline (Woodway ELG 90, Weil am Rhein, Germany). During MICE, participants performed a warm-up for 10 min at a self-selected intensity, followed by a 5 min break. Participants then ran for 50 min at 70 % of their $\dot{V}O_{2peak}$. During HIIE, participants performed 7 min of warm-up and cool-down at 70% $\dot{V}O_{2peak}$ with six bouts of high-intensity running at 90 % $\dot{V}O_{2peak}$ in between. Each high-intensity bout lasted 3 min, followed by 3 min of active recovery at 50 % $\dot{V}O_{2peak}$. Of note, although our MICE protocol was originally described as "moderate-intensity exercise"[58,59], current consensus statements on physical activity and exercise intensity terminology classify exercise at 70 % $\dot{V}O_{2peak}$ as high-intensity exercise[60]. However, the continuous character and comparatively long duration of our MICE protocol (50 min), suggests that exercise was performed below metabolic threshold 2, i.e., below maximal lactate steady state. We have thus chosen to adopt the term "moderate-intensity exercise" as originally described by Bartlett and colleagues[58,59].

## Blood collection and sample preparation

Blood was drawn from a median antecubital vein in the supine position at baseline, immediately after exercise, and 1 h after exercise for the HIIE and MICE session, respectively. The timing of blood sampling was selected to capture standardized resting conditions (baseline), effects observable directly post exercise (immediately after exercise), and to build upon work previously published by Contrepois et al.[4], in which PBMC transcriptomics indicated a strong response between 2 and 60 min after acute exercise (1 h after exercise). Each blood draw consisted of 24 ml of whole blood collected in EDTA tubes (Vacutainer, BD). After the last blood draw, PBMCs were isolated via density gradient centrifugation. To achieve this, whole blood was first diluted with phosphate-buffered saline (PBS) and then carefully layered on top of a lymphocyte separation medium (Cytiva Ficoll-Paque™ PLUS, Fisher Scientific). After centrifugation for 30 min at room temperature and 800 g$^{-1}$, the PBMC-containing interphase was collected, washed with PBS, and centrifuged again for 10 min at room temperature and 800 g$^{-1}$. PBMCs were then resuspended in freezing medium (Recovery™ cell culture freezing medium, Thermo Fisher Scientific) and stored at −80 °C before being transferred to a −150 °C freezer on the next day until further analysis.

## Flow cytometry

**Sample preparation and data acquisition.** Flow cytometry analysis was performed using a Cytek® Aurora full-spectrum flow cytometer (Cytek Biosciences, California, USA). Cryopreserved PBMCs were

gently thawed in a water bath at 37 °C with a mean recovery of 81.28 % viable cells assessed with the Zombie NIR™ Fixable Viability Kit (Bio-Legend, San Diego, CA, USA). After incubating $1 \times 10^6$ PBMCs in 2.5 µg Fc block for 10 min at room temperature, cells were stained with anti-CD3 (BUV395, clone SK7), anti-CD4 (PerCP, clone SK3), anti-CD8 (BV750, clone SK1), anti-CD16 (PE-Cy7, clone 3G8), anti-CD25 (BUV805, clone M-A251), anti-CD56 (BUV563, clone NCAM16.2), anti-CD20 (APC, clone L27), and anti-CD19 (BV480, clone SJ25C1) antibodies (all from BD Biosciences, NJ, USA). In brief, cells were incubated in the dark with a master mix containing Brilliant Stain buffer (BD Biosciences) and antibodies against surface antigens for 30 min at 4 °C. After washing with FACS buffer, the BD Pharmingen™ Transcription Factor Buffer Set was used, and cells were fixed for 40 min at 4 °C in the dark. Thereafter, intracellular staining was done by incubating cells with an anti-Foxp3 antibody (PE, clone 259D/C7) for 45 min at 4 °C in the dark. After washing, cells were resuspended in FACS buffer and acquired on the flow cytometer within 2 h after finishing the staining protocol.

**Data processing.** Gating was performed using FlowJo™ 10.10.0 (Fig. 1C). B cells were phenotyped as $CD3^-CD56^-CD19^+CD20^+$, Natural Killer T (NKT) cells as $CD3^+CD56^+$, Natural Killer (NK) cells either as $CD56^{bright}CD16^-$ ($NK^{bright}$) or $CD56^{dim}CD16^+$ ($NK^{dim}$), and regulatory T cells ($T_{regs}$) as $CD4^+CD25^+Foxp3^+$. The person analyzing the samples was blinded to the participants' group allocation. Analysis of total blood cell counts was performed from EDTA blood using a hematology analyzer (SYSMEX XP-300, Norderstedt, Germany). The lymphocyte count was then used to calculate the absolute number of peripherally circulating lymphocyte subsets according to the cell proportions derived by flow cytometry.

### LC-MS/MS-based untargeted proteomics
Samples were processed and measured in a block randomized order[61] to prevent any technical bias that might occur during sample preparation or LC-MS/MS measurement.

**Sample preparation.** Isolated PBMCs were lysed in a RIPA buffer containing 10 mM sodium fluoride, 1 mM sodium orthovanadate, cOmplete™ Protease Inhibitor Cocktail (Merck KGaA), PhosSTOP™ (Merck KGaA), 250 µ ml⁻¹ benzonase, and 10 µ ml⁻¹ DNase I. Samples were incubated on ice for 1 h and then centrifuged at 4 °C and $13,000 \times g^{-1}$ for 15 min. Protein concentration was determined in the supernatant with a BCA assay. An amount of 10 µg of protein per sample was digested (Trypsin) using an AssayMAP Bravo liquid handling system (Agilent Technologies) running the autoSP3 protocol[62]. After sample preparation, the remaining peptides were vacuum dried and stored at − 20 °C until LC-MS/MS analysis.

**MS method Orbitrap Exploris 480.** The dried peptide sample was reconstituted (97.4 % Water, 2.5 % Hexafluoro-2-propanol and 0.1 % trifluoroacetic acid (TFA)), and 10 % of the sample were used. The LC-MS/MS analysis was carried out on an Ultimate 3000 UPLC system (Thermo Fisher Scientific) directly connected to an Orbitrap Exploris 480 mass spectrometer for a total of 120 min. Peptides were online desalted on a trapping cartridge (Acclaim PepMap300 C18, 5 µm, 300 Å wide pore; Thermo Fisher Scientific) for 3 min using 30 µl/min flow of 0.1 % TFA in water. The analytical multistep gradient (300 nL/min) was performed using a nanoEase MZ Peptide analytical column (300 Å, 1.7 µm, 75 µm x 200 mm, Waters) using solvent A (0.1% formic acid in water) and solvent B (0.1% formic acid in acetonitrile). For 102 min the concentration of B was linearly ramped from 4 % to 30%, followed by a quick ramp to 78%, after two min the concentration of B was lowered to 2% and a 10 min equilibration step appended. Eluting peptides were analyzed in the mass spectrometer using data-independent acquisition (DIA) mode. A full scan at 120 k resolution (380–1400 m/z, 300% AGC target, 45 ms maxIT) was followed 47 DIA windows. The DIA acquisition covered a mass range of 400–1000 m/z using windows of a variable width with 1 m/z overlap, an AGC target of 1000% with a maxIT set to 54 ms and recorded at a resolution of 30k. Each sample was followed by a wash run (40 min) to avoid carry-over between samples. Instrument performance and suitability was monitored by regular (approx. one per 48 h) injections of a standard sample and an in-house shiny application over the whole timeline of the experiment.

**Data analysis.** Analysis of DIA RAW files was performed with Spectronaut (Biognosys, version 19.1.240724.62635)[63] in directDIA + (deep) library-free mode. Default settings were applied with the following adaptions. Within DIA Analysis under Identification, the Precursor PEP Cutoff was set to 0.01, the Protein Qvalue Cutoff (Run) set to 0.01, and the Protein PEP Cutoff set to 0.01. In Quantification, the Proteotypicity Filter was set to Only Protein Group Specific, the Protein LFQ Method was set to MaxLFQ and the quantification window was set to Not Synchronized (SN 17). The data was searched against the human proteome from Uniprot (human reference database with one protein sequence per gene, containing 20,597 unique entries from the ninth of February 2024) and the contaminants FASTA from MaxQuant (246 unique entries from the twenty-second of December 2022).

**Data processing.** Before further analysis, the obtained dataset was checked for proteins that were identified more than once. For these duplicate results, the event with the highest number of identified precursors across all samples was kept, and all other events were deleted from the dataset. The data was then filtered for proteins that were quantified in ≥ 70 % of the samples in at least one exercise condition and measurement timepoint (i.e., HIIE/MICE baseline, post exercise, 1 h post exercise). Subsequently, we imputed the data separated by exercise condition and measurement time point using the missForest package[64]. Ultimately, proteins were annotated to match the gene names provided in the org.Hs.eg.db package for subsequent Gene Ontology (GO) analysis. Translation between gene names and Entrez gene identifiers was accomplished with the bitr function from the ClusterProfiler package[29,65].

### Quantification and statistical analysis
Samples from a total of 23 participants were available for statistical analyses. For one participant, there was no sample from 1 h after MICE due to difficulties during PBMC isolation. Statistical analysis and visualization were performed in R. If not otherwise noted, data wrangling was achieved using the dplyr[66] and tidyr[67] package and subsequently visualized with ggplot2[68] and ggpubr[69].

**Unsupervised immune cell clustering using self-organizing maps.** Flow cytometry data of each sample was cleaned using the FlowAI plugin (v3.2.3) in FlowJo 10.10.0. The remaining events were gated as described above, and live cells were downsampled to 3000 events per sample using the DownSample plugin (v3.3.1). Subsequently, downsampled events were concatenated to obtain an overall dataset containing all exercise conditions (HIIE, MICE) and measurement timepoints (baseline, post- exercise, 1 h post-exercise). This dataset was then used to perform unsupervised immune cell clustering using self-organizing maps (SOM) with the FlowSOM plugin (v4.1.0). The resulting 6 clusters were identified as CD4 + T cells, CD8 + T cells, NKT cells, $CD56^{dim}$ cells, $CD56^{bright}$ cells and B cells in the build-in Cluster Explorer in FlowJo™ 10.10.0. The overall dataset was visualized using Uniform Manifold Approximation and Projection (UMAP) via the UMAP plugin (v4.1.1), and FlowSOM clusters were superimposed via color-coding. Overall, color-coded immune cell clusters were used as a template map for subsequent clustering per exercise condition and measurement time point. To achieve this, the downsampled events were concatenated for baseline, post-exercise and 1 h post-exercise in

HIIE and MICE, respectively. Ultimately, FlowSOM clustering and UMAP were performed on each of these concatenated dataframes by applying them on the previously generated map.

**Exercise-induced mobilization of immune cells.** Exercise-induced alterations in cell counts of ten different immune cell populations (Supplementary Fig. S3) were analyzed by applying linear mixed models to the flow cytometry results ($N = 22$). Measurement timepoint and exercise condition were implemented as fixed effects, and participant ID as random effect using the lmer function from the lme4 package[70]. Results of the linear mixed models were then analyzed for time and time × condition interaction effects via analyses of variance (ANOVA) with the built-in ANOVA function from R stats without correcting for testing of multiple immune cell populations. In case of significant results, pairwise comparisons of measurement timepoints and/or exercise conditions were performed by applying the emmeans function from the emmeans package. *P*-values of pairwise comparisons were Bonferroni-corrected for multiple testing. Immune cell proportions were analyzed in the same manner (Supplementary Data S4).

**Measures of variability.** All measures of variability were calculated with unimputed data to avoid potential bias arising from imputation. Inter-individual variability was calculated as the coefficient of variation (CV) for each protein across all participants separated by exercise condition (HIIE, MICE), and measurement timepoint (baseline, post exercise, 1 h post exercise). CVs were calculated as the ratio of the standard deviation $\sigma$ to the mean $\mu$. Intra-individual variability was assessed by comparing the baseline values of the two intervention days. Relative differences between day 1 and day 2 were calculated (in percent) for each protein separated by study participant. Proteomic variability was quantified for each protein by calculating (i) the mean CV across all participants in HIIE and MICE at baseline, (ii) the mean CV across all participants in HIIE and MICE post exercise and 1 h post exercise, and (iii) the mean difference between the two baselines across all participants. Proteomic variability was also calculated, separated by exercise conditions and measurement timepoints (see Supplementary Fig. S2A, B).

**Principal component analysis.** Principal component analysis (PCA) was carried out using the built-in prcomp function from R stats. All samples were plotted with the fviz_pca_ind function from the factoextra package. Exercise condition, measurement timepoint, intervention day, and sex were used as metadata to color-code PCA results. PCAs were also computed on datasets separated by exercise condition or measurement timepoint to visually assess the impact of these variables on each other (see Fig. S2C–E).

**Linear mixed models to identify proteins altered by HIIE and/ or MICE.** To identify proteins altered by HIIE and/or MICE, a linear mixed model was fitted on the log$_2$-transformed, normalized, and imputed protein intensities via the limma R package[71]. Intra-individual correlation was estimated via the duplicateCorrelation function. The model included the exercise condition (HIIE, MICE), the measurement timepoint (baseline, post-exercise, 1 h post-exercise), and the interaction between both as fixed factors. A moderated $t$ statistic[72] was obtained for each contrast of interest via the eBayes function with estimated variance trend and robustification. The resulting *p*-values for each contrast were adjusted with the Benjamini-Hochberg procedure[73] to control the false discovery rate, and significance was declared at the adjusted 5% two-sided level. The model was subsequently extended to include sex and all two-way interactions.

**Gene ontology (GO) over-representation analysis.** Time effects of the statistical analysis with limma[71] were used to map proteins that

were significantly altered by HIIE and MICE to GO terms, respectively. GO over-representation analysis was performed with the ClusterProfiler package[29,65]. For HIIE and MICE, significantly altered proteins were compared with the entire dataset of quantified proteins, applying Benjamini-Hochberg correction of *p*-values with a *p*-value cutoff of 0.05 and a *q*-value cutoff of 0.2.

**Fuzzy c-means clustering.** Fuzzy c-means clustering was performed with the Mfuzz package[74,75]. Data was standardized using the standardize function, and the optimal number of clusters was determined by calculating the minimum centroid distance for a range of cluster numbers using the Dmin function. The optimal fuzzifier was identified with the mestamiate function.

**Biological theme comparison.** Biological theme comparison was carried out using the compareCluster function from the ClusterProfiler package[29,65]. Entrez gene identifiers of the proteins contained in the identified clusters were used as input with the function command set to "enrichGO". Benjamini-Hochberg correction was applied to *p*-values with a cutoff of 0.05 and the minimum gene set size was set to 10. The results were simplified via the simplify function using a cutoff of 0.7 and visualized separated by ontology with the cnetplot function from the enrichplot package[76].

**Gene ontology (GO) gene set enrichment analysis.** Gene set enrichment analysis was performed using the gseGO function from the ClusterProfiler package[29,65]. Entrez gene identifiers and fold changes from baseline of the proteins contained in the identified clusters were used as input, with the minimum gene set size set to 10. In case fold changes were only positive or negative, the "scoreType" command was set to "pos" or "neg", respectively. *P*-values were corrected using the Benjamini-Hochberg procedure with a *p*-value cutoff of 0.05. The underlying proteins mapping to each significant GO term were identified using the select function from the AnnotationDbi package. Shared and unique GO terms across the identified clusters were visualized with the UpSetR package[77].

**Immunoproteomic signature**
**Preselection of features.** To identify features with high association to $\dot{V}O_{2peak}$, we conducted a preselection in Python (v.3.9)[78]. The features were standardized using z transformation and included the average of mass spectrometry-based proteomics data and flow cytometry-based immunophenotyping data at baseline of intervention day 1 and 2, as well as sex, height, weight and BMI. $\dot{V}O_{2peak}$ was scaled to body weight. Data from 2 participants were excluded from the analysis due to incomplete feature sets. Pairwise Spearman's rank correlations between all features and $\dot{V}O_{2peak}$ were calculated (Supplementary Data S10), and features with a correlation coefficient of > 0.4 or < − 0.4 were included in the subsequent analysis. From a total of 6063 initial features, 260 remained after this selection.

$\dot{V}O_{2peak}$ **prediction.** We ran LASSO[79,80] and ridge regression[81] as well as a random forest[82] as a non-linear, tree-based approach. A leave-one-out (LOO) cross-validation was performed in Python (v.3.9) to assess the predictive performance of these methods based on the 260 features. To optimize the hyperparameters for each model by grid search, a second inner cross-validation was performed. For each training set, we selected the model that had the lowest test error. The predicted output value resulted from the cross-validation iteration, where the corresponding output data point and its associated features were not included in the training set. These predicted values were used to calculate the mean squared error (MSE) and the r-squared. Ridge regression outperformed the other models. To preclude that our prediction model was affected from overfitting, we reran the model with permuted outcome variables to simulate data without any true

signal. The resulting model yielded an R-squared of 0.0062 and an MSE of 36.4, suggesting that the predictive performance under random conditions is close to zero. This comparison supports the conclusion that our original model does not appear to suffer from overfitting. All features with coefficients from the ridge regression are listed in the supplementary information (Supplementary Data S10).

**Immunoproteomic signature.** The immunoproteomic signature was defined as the 10 features displaying the most positive ridge regression coefficients and the 10 features displaying the most negative ridge regression coefficients in $\dot{V}O_{2peak}$ prediction. Proteins included in the immunoproteomic signature were used to create a weighted, undirected network using Spearman's rank correlations. The network was visualized in R (v.4.4.1) with the packages Hmisc (v.5.2.1) and igraph (v.2.1.1.). The immunoproteomic signature score for each participant $i$ was calculated as the weighted sum of the 20 z-transformed protein abundances ($z_p$) using the corresponding average beta coefficients ($\bar{\beta}_p$) from ridge regression models:

$$\text{Immunoproteomic signature score}_i = \sum_{p=1}^{20} \bar{\beta}_p \bullet z_{i,p}$$

**Variance partitioning.** Variance partitioning was performed using the variancePartition package in R[83]. $\dot{V}O_{2peak}$, BMI and age were included as fixed effects and participant ID and sex were included as random effects in the linear mixed models. Variance partitioning was performed on three different feature sets containing either all flow cytometry and LC-MS/MS data, the 260 preselected features from $\dot{V}O_{2peak}$ prediction or the proteins contained in the immunoproteomic signature. To compare the variance explained by each participant characteristic to the variance explained by $\dot{V}O_{2peak}$, two-sided pairwise Wilcoxon rank-sum tests with Bonferroni correction were conducted. The variance explained by participant characteristics was then assessed for each protein contained in the immunoproteomic signature. In addition, the relative contribution of participant characteristics to the explained variance of each protein was calculated.

**Correlation analyses.** Correlation analyses were performed for the association between immunoproteomic signature scores and $\dot{V}O_{2peak}$ and NAMPT z-scores and $\dot{V}O_{2peak}$. Pearson correlation coefficients were calculated and reported with corresponding two-sided $p$-values. A median split of our study cohort based on $\dot{V}O_{2peak}$ was performed (median $\dot{V}O_{2peak} = 55.6$ ml min$^{-1}$ kg$^{-1}$) and used to compare immunoproteomic signature scores between participants with a cardiorespiratory fitness above vs. below the median. Cardiorespiratory fitness groups were compared using Welch's two-sample $t$ tests.

### Reporting summary
Further information on research design is available in the Nature Portfolio Reporting Summary linked to this article.

## Data availability
All data associated with this article can be explored via our interactive web application at https://sportsmedicine-dortmund.shinyapps.io/beat. Raw data files of all samples processed in the proteomics analysis are hosted on the PRoteomics IDEntifications Database (PRIDE) under the following URL: https://www.ebi.ac.uk/pride/archive/projects/PXD058573. Raw data files of all samples processed in the flow cytometry analysis are hosted on https://figshare.com under the following URL: https://doi.org/10.6084/m9.figshare.30543317. To ensure reproducibility of our analysis, allocation of raw data files to study participants is provided in Supplementary Data S1. All figures were created from the provided raw data using the indicated software, R or python packages.

## Code availability
No custom code was generated for this analysis. All applied analysis tools are specified in the methods section of the article.

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

## Acknowledgements

We want to thank Lars Donath and Ludwig Rappelt for their help in conducting the trial. We thank the team of the Proteomics Core Facility of the DKFZ, particularly Adrian Stoegbauer and Alina Ertl for sample preparation and LC-MS/MS measurement. We also want to thank the German Sport University Cologne for supplying internal funds to A.J.M. Schematic figures were created with https://BioRender.com.

## Author contributions

Conceptualization, N.J., A.J.M., and P.Z.; Methodology, N.J., A.J.M., A.S., and P.Z.; Software, D.W., C.We., M.S., and S.C.; Formal Analysis, D.W., S.P., C.We., M.S., and S.C.; Investigation, A.J.M., S.P., A.S., M.S., and D.H.; Resources, C.Wa., C.A.O., D.H., and P.Z.; Writing – Original Draft, D.W., N.J., S.P., C.We., M.S., and S.C.; Writing – Review & Editing, A.J.M., S.P., A.S., C.We., A.L.H., M.S., S.C., A.G, C.Wa., C.A.O., D.H., and P.Z.; Visualization, D.W. and C.We.; Supervision, P.Z., A.G., and D.H., Project Administration, P.Z.; Funding Acquisition, A.J.M.

## Funding

## Competing interests

The authors declare no competing interests.
