## [Transparent Peer Review file · Nature Communications]

Acute exercise rewires the proteomic landscape of human immune cells

Corresponding Author: Professor Philipp Zimmer

Version 0:

Reviewer comments:

Reviewer #1

(Remarks to the Author)

This noteworthy manuscript reports findings of a translational study of the effects of aerobic exercise intensity on immune cell proteomic profiles in younger, healthy, physically fit persons. In a randomized, cross-over design, participants completed treadmill running acute bouts of high intensity interval exercise or moderate intensity continuous state exercise and had blood drawn for peripheral blood mononuclear cell (PBMC) isolation and storage pre-exercise, post-exercise, and post-one hour recovery. Stored PBMC were then used for bulk proteomic analyses. The main findings of this study are that high intensity exercise—compared to moderate intensity—leads to greater proteomic changes in PBMCs linked to immune response, independent of changes in specific immune cell concentrations after exercise. The authors also identified baseline PBMC profiles that are highly associated with and predictive of cardiorespiratory fitness. The strengths of this manuscript are the clear and concise writing, reporting, and data presentation, use of human samples, robust study design and statistical analyses plan, and novel analyses and findings of exercise-induced immune cell-specific proteomic changes. The weaknesses of the manuscript include lack of data integration/analyses of with other level -omics (e.g., transcriptomics, epigenetics) or immune function assessments, and some lack of clarity regarding the goals and hypotheses of this exploratory project. Overall, the novel findings of this manuscript considerably add to the field of exercise immunology. The study holds promise to inspire future work aimed at understanding the effects of exercise on immune function, which may have significant translational impact in the health sciences and clinical care. My comments for suggested minor revisions are as follows:

1. The Introduction and background leading to the present study are well presented. However, the rationale and goals of the project are minimally described. The addition of a more detailed rationale for study, including hypotheses regarding possible differential effects of high vs moderate intensity exercise, would strengthen the manuscript.
2. Though implied, the rationale to investigate bulk vs targeted proteomics is not clearly discussed. Please specify in the Introduction and/or Methods.
3. The findings of strong relationships between baseline PBMC proteomics and cardiorespiratory fitness and possible predictive value are fascinating. However, there is minimal rationale for why these analyses are included in the current study and how these analyses fit within the context of the rest of the analyses discussed (i.e., focus on acute bout changes). Please discuss.
4. The findings of this study give support to a concept that exercise boosts immune responses to pathogens and shows differential effects of high versus moderate intensity exercise. Consider adding further comment in the Discussion regarding how findings of the study add to current understanding of how acute exercise modulates infection risk (e.g., compared to previous notion that prolonged, high intensity exercise may be immunosuppressive).
5. The findings of differential protein effects of time and group focus on immune effector function. Consider adding further discussion on other biologic processes impacted in PBMCs after acute exercise. For example, it is interesting that cluster 4 proteins—which decreased at both post-exercise and recovery time points—strongly associated with mitochondrial/cellular metabolism pathways, yet VO₂ peak was linked to greater baseline PBMC mitochondrial/metabolic protein abundance.
6. It may also be helpful to add discussion for what next steps are needed to better understand and apply the findings of the study.
7. In the Discussion, recommend clarifying that high intensity aerobic exercise was needed to stimulate changes in PBMC effector function specifically in healthy, young, trained individuals (i.e., findings may be evidence of a training or age effect, as it is still unclear if moderate intensity may produce a robust response in older, sedentary, untrained, and chronic disease populations).

8. The addition of a "limitations" paragraph is needed to strengthen the Discussion. For example, as above, it is unclear if findings apply to older, sedentary, or chronic disease populations; there is lack of integration of proteomic data with transcriptomic or epigenetic data to better understand molecular signaling pathways; there is no connection of proteomic data with immune function assessments or immunometabolic assessments.
9. In the Methods, some clarification of experimental procedures is needed. For example, 70% of V02 peak can be considered to steady state high-intensity (as opposed to moderate intensity as described in the manuscript). Please provide a brief clarification and rationale.
10. Please also provide brief rationale for acute exercise bout duration.
11. Please provide brief rationale for the timing of blood sample collection. For example, why inclusion of 1 hour post-acute bout compared to a different time point (e.g., 30 minutes, 4 hours, 24 hours).

Reviewer #2

(Remarks to the Author)

This study presents a rigorous and thoughtfully designed investigation into how acute aerobic exercise shapes the molecular landscape of peripheral blood mononuclear cells (PBMCs), filling a critical gap in the field of exercise immunology from a proteomics perspective. By employing a randomized crossover design, carefully matched for workload and duration, and combining high-resolution LC-MS/MS proteomics with spectral flow cytometry, the authors provide a solid experimental foundation. Notably, the discovery that high-intensity interval exercise (HIIE) induces a much stronger proteomic response than moderate-intensity continuous exercise (MICE) adds biological weight to WHO's recommendations highlighting the importance of exercise intensity for health promotion.

Beyond its depth of proteomic analysis, the study stands out for its integration of functional annotation through Gene Ontology, hierarchical and fuzzy clustering, and its correlation with cardiorespiratory fitness (VO₂peak). The identification of NAMPT and other proteins related to metabolism and immune function as potential predictors of VO₂peak is particularly exhilarating, offering new avenues to understand how immunometabolic health relates to systemic fitness.

That said, a few refinements could further enhance the clarity, depth, and translational relevance of this important work:

1. Streamlining the narrative: Some parts of the text are quite dense. Condensing and restructuring these sections could make the key findings easier to grasp and visually more accessible for the reader.
 2. Incorporating sex- and age-based comparisons: Including analyses based on sex and age could offer valuable insights, especially given the well-established hormonal and metabolic differences across sexes and life stages. Such stratification may reveal distinct molecular responses to exercise and aid in tailoring interventions.
 3. Linking molecular pathways to clinical relevance: A brief discussion on how the activated pathways might be beneficial in metabolic conditions such as type 2 diabetes or how they might protect organs from chronic disease progression would elevate the translational impact of the findings and support differential applications of MICE and HIIE.
 4. Adding functional validation: Although functional predictions were made using Gene Ontology, no direct functional assays (e.g., cytokine secretion, cytotoxicity, or antigen presentation) were performed. Including or proposing these validations, particularly for key markers like NAMPT, would strengthen confidence in the interpretations and reduce the risk of overfitting.
 5. Integrating additional omics layers: While the correlation of proteomic changes with VO₂peak is valuable, incorporating transcriptomic, metabolomic, or cytokine data would have provided a more holistic systems biology view. Cytokine profiling, in particular, would have been a strong complement to the immune-related findings.
 6. Quantifying effect sizes: The observation that HIIE induces stronger changes could be bolstered by reporting effect sizes, which would add statistical weight and practical context to the biological findings.
 7. Website access issue: The resource link (<https://sportsmedicine-dortmund.shinyapps.io/beat>) appears to require a password and was not functional with the provided materials.
- Addressing these points would help the study not only read more clearly but also broaden its impact—both for the scientific community and in real-world health and exercise applications.

Reviewer #3

(Remarks to the Author)

I had the opportunity to review the manuscript "Acute exercise rewires the proteomic landscape of human immune cells" by D. Walzik et al. The manuscript describes results of a randomized crossover trial comparing effects of high-intensity interval exercise and of moderate-intensity continuous exercise on proteomic makeup of PBMCs. I reviewed the manuscript with a focus on the statistical methods used, but I am not an expert in the analysis of high-dimensional OMICS data and am not fully familiar with the statistical conventions in that field. I have a few concerns about the methods and would ask authors to comment on them.

Linear mixed effects models were used for analysis of immune cells and for proteins. The analytical approaches differ in some certain aspects, as e.g. the adjustment for multiple testing. The authors should comment on the differences in the approaches and it should also be clear in the results sections and in the figures, which approach was used. For the immune cells, it is unclear which factor was used for the Bonferroni adjustment - was the correction applied for multiple tests over different timepoints or was the number of tested cells also considered?

Was the randomization sequence considered in the regression model or was it used in the analysis in any other way?

For the prediction model presented in the last part of the results, I would be really interested how this model performs compared to a model derived from data of the same structure without any true signal - which can be generated e.g. by

permuting the outcome variable. This would allow to assess how much the resulting R-squared is affected from overfitting.

Generally, I believe that some of the wording used in the manuscript implies a causality that cannot be confirmed by the data ("... rules out the possibility ...", "... that were ... altered by ...", etc.). The authors should carefully review the manuscript in this regard and adjust the wording accordingly.

Version 1:

Reviewer comments:

Reviewer #1

(Remarks to the Author)

Thank you for your detailed revisions and responses. All of my comments have been adequately addressed.

Reviewer #2

(Remarks to the Author)

This manuscript presents a well-designed and carefully executed investigation into how acute aerobic exercise influences the proteomic landscape of PBMCs. The combination of a randomized crossover design, controlled exercise workloads, and high-resolution LC-MS/MS proteomics provides a solid methodological foundation. The analytical approaches—ranging from Gene Ontology enrichment to clustering analyses and correlations with $\dot{V}O_{2peak}$ —are thoughtfully applied and add depth to the interpretation. Notably, the finding that high-intensity interval exercise elicits a more pronounced proteomic response than moderate-intensity exercise is convincing, and the additional analyses involving NAMPT and variance partitioning strengthen the study's connection to cardiorespiratory fitness.

The authors have responded well to prior reviewer comments by clarifying dense narrative sections, addressing sex- and age-related considerations, expanding statistical reporting, and improving translational context. The manuscript is clear, scientifically rigorous, and supported by appropriate methodological detail. Although some results sections could still benefit from slight condensation for readability, and future work incorporating functional assays or multi-omics layers would enrich mechanistic insight, these points do not detract from the quality of the present study. Overall, the work makes a meaningful contribution to exercise immunology and is well-positioned for publication.

Reviewer #3

(Remarks to the Author)

I would like to thank the authors for taking my comments into consideration and for their clarifications.

Point by point response

Dear Reviewers,

Thank you very much for your constructive and positive feedback on our submission. Please find the point-by-point response addressing your comments below.

We believe that the revision process and your input have improved our manuscript and we are looking forward to receiving feedback.

Best regards on behalf of our team,

Philipp Zimmer

Reviewer #1 (Remarks to the Author):

This noteworthy manuscript reports findings of a translational study of the effects of aerobic exercise intensity on immune cell proteomic profiles in younger, healthy, physically fit persons. In a randomized, cross-over design, participants completed treadmill running acute bouts of high intensity interval exercise or moderate intensity continuous state exercise and had blood drawn for peripheral blood mononuclear cell (PBMC) isolation and storage pre-exercise, post-exercise, and post-one hour recovery. Stored PBMC were then used for bulk proteomic analyses. The main findings of this study are that high intensity exercise—compared to moderate intensity—leads to greater proteomic changes in PBMCs linked to immune response, independent of changes in specific immune cell concentrations after exercise. The authors also identified baseline PBMC profiles that are highly associated with and predictive of cardiorespiratory fitness. The strengths of this manuscript are the clear and concise writing, reporting, and data presentation, use of human samples, robust study design and statistical analyses plan, and novel analyses and findings of exercise-induced immune cell-specific proteomic changes. The weaknesses of the manuscript include lack of data integration/analyses of with other level -omics (e.g., transcriptomics, epigenetics) or immune function assessments, and some lack of clarity regarding the goals and hypotheses of this exploratory project. Overall, the novel findings of this manuscript considerably add to the field of exercise immunology. The study holds promise to inspire future work aimed at understanding the effects of exercise on immune function, which may have significant translational impact in the health sciences and clinical care. My comments for suggested minor revisions are as follows:

Thank you for reviewing our manuscript and providing helpful and constructive comments. We fully agree with your comments and revised our manuscript accordingly.

1. The Introduction and background leading to the present study are well presented. However, the rationale and goals of the project are minimally described. The addition a more detailed rationale for study, including hypotheses regarding possible differential effects of high vs moderate intensity exercise, would strengthen the manuscript.

We added a clear rationale including a hypothesis on the possible differential effects of HIIE vs. MICE at the end of the introduction section and expanded the first sentence of the abstract by a rationale for doing proteomic analyses on PBMCs.

2. Though implied, the rationale to investigate bulk vs targeted proteomics is not clearly discussed. Please specify in the Introduction and/or Methods.

We added the rationale and an explanation for investigating bulk proteomics in the introduction section.

3. The findings of strong relationships between baseline PBMC proteomics and cardiorespiratory fitness and possible predictive value are fascinating. However, there is minimal rationale for why these analyses are included in the current study and how these analyses fit within the context of the rest of the analyses discussed (i.e., focus on acute bout changes). Please discuss.

Thank you for drawing attention to the lack of rationale regarding the relationship between baseline proteomics and cardiorespiratory fitness. We added further pieces of information in the abstract and the introduction section and elaborated on the reasons for doing this analysis in the results and discussion sections.

4. The findings of this study give support to a concept that exercise boosts immune responses to pathogens and shows differential effects of high versus moderate intensity exercise.

Consider adding further comment in the Discussion regarding how findings of the study add to current understanding of how acute exercise modulates infection risk (e.g., compared to previous notion that prolonged, high intensity exercise may be immunosuppressive)

As suggested, we added a paragraph in the discussion section on the question how acute exercise modulates infection risk and thus further contextualized our findings.

5. The findings of differential protein effects of time and group focus on immune effector function. Consider adding further discussion on other biologic processes impacted in PMBCs after acute exercise. For example, it is interesting that cluster 4 proteins—which decreased at both post-exercise and recovery time points—strongly associated with mitochondrial/cellular metabolism pathways, yet VO₂ peak was linked to greater baseline PBMC mitochondrial/metabolic protein abundance.

Thank you for suggesting this. We have highlighted the differential response of mitochondrial/metabolic proteins (Figure S3 vs. Figure 5B) in the discussion.

6. It may also be helpful to add discussion for what next steps are needed to better understand and apply the findings of the study.

At the end of the discussion section, we added an outlook paragraph on the future methodological steps that need to be taken.

7. In the Discussion, recommend clarifying that high intensity aerobic exercise was needed to stimulate changes in PBMC effector function specifically in healthy, young, trained individuals (i.e., findings may be evidence of a training or age effect, as it is still unclear if moderate intensity may produce a robust response in older, sedentary, untrained, and chronic disease populations).

As suggested, we clarified in the discussion/limitations that our findings only apply to young health trained individuals, and that future studies in other cohorts are needed for further validation.

8. The addition of a “limitations” paragraph is needed to strengthen the Discussion. For example, as above, it is unclear if findings apply to older, sedentary, or chronic disease populations; there is lack of integration of proteomic data with transcriptomic or epigenetic data to better understand molecular signaling pathways; there is no connection of proteomic data with immune function assessments or immunometabolic assessments.

A paragraph on the limitations including the limited generalizability of the findings to other populations and the lack of other omics layers (also in line with comment 5 from Reviewer#2) and immune function assessments was added at the end of the discussion section.

9. In the Methods, some clarification of experimental procedures is needed. For example, 70% of VO₂ peak can be considered to steady state high-intensity (as opposed to moderate intensity as described in the manuscript). Please provide a brief clarification and rationale.

Thank you for this comment. We acknowledge that there are some discrepancies in exercise science literature concerning the terminology used for intensity domains. We based our terminology on the original articles that described the time- and energy-matching between exercise sessions performed at different intensities (PMIDs 22267390 and 21360405). In these articles the authors use the terms “high-intensity exercise” and “moderate-intensity exercise” to distinguish between a 6 x 3-min HIT protocol at 90% VO₂max and a 50-min continuous running protocol at 70% VO₂max.

We agree that the 70% $\dot{V}O_{2peak}$ used in study can be considered high-intensity exercise, as defined by the joint American College of Sports Medicine (ACSM) Expert Statement and Exercise and Sport Science Australia (ESSA) Consensus Statement (PMID 41085254). However, the same consensus statement also defines moderate intensity as exercise performed between metabolic threshold (MT)1 and MT2. The continuous character and comparatively long duration of our moderate-intensity exercise protocol (50 min) implies that exercise was performed below metabolic threshold 2, i.e., below maximal lactate steady state. Conversely, exercise above MT2 – which would be referred to as high intensity exercise according to the consensus statement – could not have been sustained for a duration of 50 min since per definition this intensity is above maximal lactate steady state. To refer to this topic in a transparent manner, we added a short section and a reference to the expert/consensus statement in the methods section to explain our use of terminology.

10. Please also provide brief rationale for acute exercise bout duration.

A rationale for the exercise bout duration was added in the methods section. We decided to use a conventional exercise duration that is often used in different exercise settings and selected a previously published protocol comparing the effects of HIIE vs. MICE.

11. Please provide brief rationale for the timing of blood sample collection. For example, why inclusion of 1 hour post-acute bout compared to a different time point (e.g., 30 minutes, 4 hours, 24 hours).

We added a brief rationale for the timing of blood sampling. Samples were collected at baseline (as standardized reference point), immediately post exercise (to capture effects most directly induced by acute exercise), and 1h post exercise (to be consistent with other works published in the field and particularly to building upon the work of Contrepois et al. 2020, Cell (PMID: 32470399), in which PBMC transcriptomics indicated a strong response in between 2-60 min. after acute exercise).

Reviewer #2 (Remarks to the Author):

This study presents a rigorous and thoughtfully designed investigation into how acute aerobic exercise shapes the molecular landscape of peripheral blood mononuclear cells (PBMCs), filling a critical gap in the field of exercise immunology from a proteomics perspective. By employing a randomized crossover design, carefully matched for workload and duration, and combining high-resolution LC-MS/MS proteomics with spectral flow cytometry, the authors provide a solid experimental foundation. Notably, the discovery that high-intensity interval exercise (HIIE) induces a much stronger proteomic response than moderate-intensity continuous exercise (MICE) adds biological weight to WHO's recommendations highlighting the importance of exercise intensity for health promotion.

Beyond its depth of proteomic analysis, the study stands out for its integration of functional annotation through Gene Ontology, hierarchical and fuzzy clustering, and its correlation with cardiorespiratory fitness ($\dot{V}O_{2peak}$). The identification of NAMPT and other proteins related to metabolism and immune function as potential predictors of $\dot{V}O_{2peak}$ is particularly exhilarating, offering new avenues to understand how immunometabolic health relates to systemic fitness. That said, a few refinements could further enhance the clarity, depth, and translational relevance of this important work:

Thank you very much for your positive and constructive feedback on our manuscript. We agree with your comments and believe that the corresponding revisions helped to further improve the quality of the manuscript.

1. Streamlining the narrative: Some parts of the text are quite dense. Condensing and restructuring these sections could make the key findings easier to grasp and visually more accessible for the reader.

This comment is partly consistent with some comments from Reviewer 1 (comment 1, 7, 8). We sharpened our rationale in the introduction, including a hypothesis on comparing the effects of HIIE vs. MICE, and further contextualized the findings of study in the discussion section. If you have any specific remarks on how to condense the results section, we are happy to adapt.

2. Incorporating sex- and age-based comparisons: Including analyses based on sex and age could offer valuable insights, especially given the well-established hormonal and metabolic differences across sexes and life stages. Such stratification may reveal distinct molecular responses to exercise and aid in tailoring interventions.

Thank you for this comment. We agree that meta-variables like sex and age can offer valuable insights into immunological differences in response to exercise. In fact, we have already evaluated sex differences in our analysis and did not find any significant interaction effects when including sex as a fixed effect in our linear mixed models (see page 7, last paragraph). Regarding age, we think our study is not suited to address this question since our recruited cohort was characterized by a very homogenous age distribution (30 ± 4 years), also because age between 18 and 35 was defined as an inclusion criterium. However, we added a paragraph at the end of the discussion that highlights the need of future studies to investigate other populations including cohort of different age and fitness.

3. Linking molecular pathways to clinical relevance: A brief discussion on how the activated pathways might be beneficial in metabolic conditions such as type 2 diabetes or how they might protect organs from chronic disease progression would elevate the translational impact of the findings and support differential applications of MICE and HIIE.

In accordance with your suggestion, we added a brief statement in the discussion section to further elevate the translational impact of our findings.

4. Adding functional validation: Although functional predictions were made using Gene Ontology, no direct functional assays (e.g., cytokine secretion, cytotoxicity, or antigen presentation) were performed. Including or proposing these validations, particularly for key markers like NAMPT, would strengthen confidence in the interpretations and reduce the risk of overfitting.

Thank you for highlighting this. With our analysis we aimed to provide an overview of proteomic alterations in PBMCs and the subsequent triggered molecular pathways. From our point of view functional/mechanistic validation of our results is much better suited for analyses that focus on specific immune cell populations. We have addressed the lack of functional assays like cytokine secretion or cytotoxicity assays in the discussion section and provided guidance for future steps to be taken in this direction.

Regarding NAMPT, we revisited the data obtained in this study to strengthen the confidence in the interpretation that higher NAMPT levels in PBMCs are observed in persons with higher cardiorespiratory fitness (i.e., VO₂peak). For this purpose, we have expanded the manuscript by the following aspects:

1. We calculated immunoproteomic signature score for each participant and included a correlation analysis and a median split of our cohort based on VO₂peak values to demonstrate that immunoproteomic signature scores correlate with VO₂peak and that this metric differs between participants with VO₂peak above the median (55.6 ml/min/kg) vs. below the median (Figure 5D).

2. We provided an internal validation that NAMPT z-scores obtained via LC-MS/MS are correlated with VO₂peak (Figure 5E).
3. We included a variance partitioning approach in Figure 5 and Figure S5 to demonstrate that:
 - a. VO₂peak accounts for the largest fraction of explainable variance in the immunoproteomic signature (Figure 5F & S5A).
 - b. from all proteins included in the immunoproteomic signature, VO₂peak provides 93.66% of the explainable variance for NAMPT (Figure 5G & S5B). This indicates that cardiorespiratory fitness contributes more to explainable variance in NAMPT than in other proteins and that other participant characteristics like sex, age or BMI play an inferior role in explaining the variance within NAMPT values.

Based on these additions to Figure 5 and Figure S5, we have provided details in the methods section for variance partitioning, calculation of immunoproteomic signature scores, and correlation analyses. We have also rewritten the results section on the immunoproteomic signature with the overall aim to strengthen the confidence in our results.

Regarding the risk of overfitting in our VO₂peak prediction (see also second last comment of Reviewer 3), we reran the prediction model with permuted outcome variables to simulate data without any true signal. The resulting model yielded an R² of 0.0062 and a mean square error (MSE) of 36.4, suggesting that the predictive performance under random conditions is close to zero. This comparison supports the conclusion that our original model does not appear to suffer from overfitting. We added a few sentences on this in the results section.

5. Integrating additional omics layers: While the correlation of proteomic changes with VO₂peak is valuable, incorporating transcriptomic, metabolomic, or cytokine data would have provided a more holistic systems biology view. Cytokine profiling, in particular, would have been a strong complement to the immune-related findings.

We fully agree with this comment and mentioned this aspect as key limitation at the end of the discussion section.

6. Quantifying effect sizes: The observation that HIIE induces stronger changes could be bolstered by reporting effect sizes, which would add statistical weight and practical context to the biological findings.

Thank you for this comment. We have added log₂ fold changes as standardized effect sizes together with FDR-adjusted p values for all time × condition interaction effects reported in the text (see section Proteomic alterations differ between HIIE and MICE). A full report of all statistical results, including all log₂ FCs is given in Table S5, which is cross-referenced several times throughout the paragraphs on interaction effects.

7. Website access issue: The resource link (<https://sportsmedicine-dortmund.shinyapps.io/beat>) appears to require a password and was not functional with the provided materials.

Thank you for letting us know about this. We had provided the editors with a password, which was apparently not forwarded to the reviewers. The website is now openly available without the need of a password (<https://sportsmedicine-dortmund.shinyapps.io/beat>). We have also added a reference to the website in the abstract to promote interactive data mining for researchers interested in our results.

Addressing these points would help the study not only read more clearly but also broaden its impact—both for the scientific community and in real-world health and exercise applications.

We thank this reviewer for the constructive review and valuable input aimed at improving our manuscript.

Reviewer #3 (Remarks to the Author):

I had the opportunity to review the manuscript "Acute exercise rewires the proteomic landscape of human immune cells" by D. Walzik et al. The manuscript describes results of a randomized crossover trial comparing effects of high-intensity interval exercise and of moderate-intensity continuous exercise on proteomic makeup of PBMCs. I reviewed the manuscript with a focus on the statistical methods used, but I am not an expert in the analysis of high-dimensional OMICS data and am not fully familiar with the statistical conventions in that field. I have a few concerns about the methods and would ask authors to comment on them.

Thank you for reviewing our manuscript and your constructive comments. Please find our replies to your comments below.

Linear mixed effects models were used for analysis of immune cells and for proteins. The analytical approaches differ in some certain aspects, as e.g. the adjustment for multiple testing. The authors should comment on the differences in the approaches and it should also be clear in the results sections and in the figures, which approach was used. For the immune cells, it is unclear which factor was used for the Bonferroni adjustment - was the correction applied for multiple tests over different timepoints or was the number of tested cells also considered?

Thank you for noting this. We acknowledge that immune cell counts and proteomics data were analyzed differently and have specified which approach was used in the results and methods section as well as the figure legends.

Since immune cell counts were only analyzed for a total of 10 immune cell populations, we refrained from correcting linear mixed model and subsequent ANOVAs for multiple testing. In contrast, pairwise comparisons of timepoints within each group and pairwise comparisons of groups within each timepoint were Bonferroni-corrected. Against the backdrop of the higher number of statistical tests for pairwise comparisons ((6 time effect + 3 interaction effects) * 10 outcome variables = 90 pairwise comparisons) compared to ANOVAs (1 ANOVA per outcome variable = 10 ANOVAs), this is how, to the best of our knowledge, correction for multiple testing is classically handled in exercise science research with a manageable amount of outcome variables.

In contrast, proteomics data comprised 6039 proteins across 23 participants in 2 conditions with 3 measurement timepoints each. For large datasets with thousands of outcome variables more sophisticated analytical pipelines with correction for multiple testing are urgently required. In consultation with our colleagues from the proteomics core facility and biostatistics department at the German Cancer Research Center, we have thus decided to implement linear mixed models with Benjamini-Hochberg correction for multiple testing using the limma R package (PMID 25605792). Limma is especially powerful for statistical analyses in complex study designs as the cross-over design applied by us.

Was the randomization sequence considered in the regression model or was it used in the analysis in any other way?

We did not include randomization sequence in the regression models for statistical analysis of immune cell counts or protein abundances. We acknowledge that if exercise sequences were not randomized, it would be crucial to include the sequence of interventions as a fixed effect in the statistical analyses. However, with randomized exercise sequences as employed by us, we do not expect any differences in immune cell counts/protein abundances. This notion is supported by the fact that randomization sequences did not differ in terms of participant characteristics (see section on Randomization in Methods and Table S1). We also performed principal component analysis (PCA) to assess the impact of randomization sequence on the variation within our data but did not include it in the manuscript since it offers little additional

information compared to the PCAs performed for intervention day (which was not randomized) and sex (Figure S2C). All three figures are appended below for your information.

For the prediction model presented in the last part of the results, I would be really interested how this model performs compared to a model derived from data of the same structure without any true signal - which can be generated e.g. by permuting the outcome variable. This would allow to assess how much the resulting R-squared is affected from overfitting.

We appreciate this great suggestion. In line with the reviewer's comment, we tested the model performance using a permuted version of the outcome variables to simulate data without a true signal. The resulting model yielded an R² of 0.0062 and a mean square error (MSE) of 36.4, suggesting that the predictive performance under random conditions is close to zero. This comparison supports the conclusion that our original model does not appear to suffer from overfitting. We added a few sentences on this in the methods section.

Generally, I believe that some of the wording used in the manuscript implies a causality that cannot be confirmed by the data ("... rules out the possibility ...", "... that were ... altered by ...", etc.). The authors should carefully review the manuscript in this regard and adjust the wording accordingly.

Thank you for this comment. We have reworded sections of our manuscript that imply false causality.